

# EFFECTS OF BRACKISH WATER INFLOW ON METHANE CYCLING MICROBIAL COMMUNITIES IN A FRESHWATER REWETTED COASTAL FEN

Cordula Nina Gutekunst[1], Susanne Liebner[2,3] Anna-Kathrina Jenner[4], Klaus-Holger Knorr[5], Viktoria Unger[6], Franziska Koebsch[7], Erwin Don Racasa[8], Sizhong Yang[2], Michael Ernst Böttcher[4,9,10], Manon Janssen[8], Jens Kallmeyer[2], Denise Otto[4], Iris Schmiedinger[4], Lucas Winski[4,11], Gerald Jurasinski[1,10]

[1]Landscape Ecology, University of Rostock, Rostock, 18059, Germany
[2]Section Geomicrobiology, German Research Centre for Geosciences (GFZ), Potsdam, 14473, Germany
[3]Institute of Biochemistry and Biology, University of Potsdam, Potsdam, 14476, Germany
[4]Geochemistry and Stable Isotope Biogeochemistry, Leibniz Institute for Baltic Sea Research (IOW), Warnemünde, 18119, Germany
[5]Institute of Landscape Ecology, Ecohydrology & Biogeochemistry Group, University of Münster, Münster, 48149, Germany
[6]Institute of Plant Science and Microbiology, Applied Plant Ecology, University of Hamburg, Hamburg, 22609, Germany
[7]Bioclimatology, University of Göttingen, Göttingen, 37073, Germany
[8]Soil Physics, University of Rostock, Rostock, 18059, Germany
[9]Marine Geochemistry, University of Greifswald, Greifswald, 17489, Germany
[10]Department of Maritime Systems, University of Rostock, Rostock, 18059, Germany
[11]Present address: FRG, University of Jena, Jena, 07743, Germany

*Correspondence to*: Cordula N. Gutekunst (cordula.gutekunst@uni-rostock.de)

**Abstract.** Rewetted peatlands can be a significant source of methane ($CH_4$), but in coastal ecosystems, input of sulfate-rich seawater could potentially mitigate these emissions. The presence of sulfate as electron acceptor during organic matter decomposition is known to suppress methanogenesis, by favoring the growth of sulfate-reducers, which outcompete methanogens for substrate. We investigated the effects of a brackish water inflow on the microbial communities relative to

$CH_4$ production-consumption dynamics in a freshwater rewetted fen at the southern Baltic Sea coast after a storm surge in January 2019 and analyzed our data in context with the previous freshwater rewetted state (2014 serves as our baseline) and the conditions after a severe drought in 2018.

We took peat cores at four previously sampled locations along a brackishness gradient to compare soil and pore water geochemistry as well as the microbial methane and sulfate cycling communities with the previous conditions. We used high-
throughput sequencing and quantitative polymerase chain reaction (qPCR) to characterize pools of DNA and cDNA targeting total and putatively active bacteria and archaea. Furthermore, we measured $CH_4$ fluxes along the gradient and determined the concentrations and isotopic signatures of trace gases in the peat.

We found that both, the inflow effect of brackish water and in parts also the preceding drought increased the sulfate availability in the surface and pore water. Still, peat soil $CH_4$ concentrations and the [13]C compositions of $CH_4$ and total dissolved inorganic
carbon (DIC) indicated ongoing methanogenesis and little methane oxidation. Accordingly, we did not observe a decrease of absolute methanogenic archaea abundance or a substantial change in methanogenic community composition following the





inflow, but found that the methanogenic community had mainly changed during the precedent drought. In contrast, absolute abundances of aerobic methanotrophic bacteria decreased back to their pre-drought level after the inflow while they had increased during the drought year. In line with the higher sulfate concentrations, the absolute abundances of sulfate reducing

bacteria (SRB) increased—as expected—by almost three orders of magnitude compared to the freshwater state and also exceeded abundances recorded during the drought by over two orders of magnitude. Against our expectations, methanotrophic archaea (ANME), capable of sulfate-mediated anaerobic methane oxidation, did not increase in abundance after the brackish water inflow. Altogether, we could find no microbial evidence for hampered methane production or increased methane consumption in the peat soil after the brackish water inflow. Because Koebsch et al. (2020) reported a new minimum in $CH_4$

fluxes at this site since rewetting of the site in 2009, methane oxidation may, however, take place in the water column above the peat soil or in the lose organic litter on the ground. This highlights the importance to consider all compartments across the peat-water-atmosphere continuum to develop an in-depth understanding of inflow events in rewetted peatlands. We propose that the changes in microbial communities and GHG fluxes relative to the previous freshwater rewetting state cannot be explained with the brackish water inflow alone, but was potentially reinforced by a biogeochemical legacy effect of the

precedent drought.

## 1 Introduction

Peatlands are important global carbon stores (Gorham, 1991; Batjes, 1996; Limpens et al., 2008; Yu et al., 2010; Page et al., 2011; Dargie et al., 2017), but drainage for agriculture or peat extraction leads to aerobic mineralization of the organic material and thus, to increased emissions of carbon dioxide ($CO_2$) (Frolking et al., 2011; Leifeld, 2013). Rewetting effectively stops the

high $CO_2$ emissions (Kirby et al., 2013; Paustian et al., 2016; Wilson et al., 2016) and can restore the carbon sink function (Leifeld and Menichetti, 2018). However, rewetting of drained peatlands may induce high emissions of methane ($CH_4$) (Joosten and Couwenberg, 2009; Wichtmann et al., 2010; Hahn et al., 2015; Abdalla et al., 2016), especially in nutrient rich fens. Although this does not negate the overall beneficial effect of peatland rewetting for mitigating climate warming (Günther et al., 2020), $CH_4$ still acts as short-lived, but strong greenhouse gas (GHG) (Lelieveld et al., 1998; Myhre et al., 2013) and thus,

high emission rates should be avoided when possible (Nisbet et al., 2020). It is therefore desirable to better understand the conditions under which $CH_4$ emissions from rewetted peatlands can be kept small to implement the best mitigation strategy.

Sea level rise, driven by global warming (Fabian, 2002; Church et al., 2013; Nerem et al., 2018) may cause a sustainable shift in the biogeochemistry of coastal wetland systems (van Dijk et al., 2019), including low lying coastal peatlands (Jurasinski et al., 2018). Above all, marine water inflow may increase the sulfate availability in these ecosystems and thereby provide an

alternative electron acceptor (EA) for organic matter (OM) decomposition (Jørgensen, 1982). Available studies typically report a reduction of methane production (methanogenesis) in anaerobic soil zones in the presence of a thermodynamically more favorable EA such as sulfate. This has been found in marine environments (Oremland, 1988), rice paddies, (van der Gon and



Neue, 1994), salt marshes (Bartlett et al., 1987) and even in freshwater peatlands (Lovley and Klug, 1983; Gauci et al., 2004;
Pester et al., 2012). Sulfate reducing bacteria (SRB) outcompete methanogens because of a higher energy gain through their

metabolic pathway (Schönheit et al., 1982; Lovley and Klug, 1983) and their high substrate affinity (Kristjansson and
Schönheit, 1983).

The majority of methanogens are obligate anaerobic methane-producing (methanogenic) archaea (Moore and Knowles, 1989;
Strack et al., 2008; Thauer et al., 2008; Nazaries et al., 2013), although they may also withstand the presence of oxygen within
anaerobic niches in oxic soil layers (Angle et al., 2017; Wagner, 2017) or tolerate short-term droughts (Kim et al., 2008; Wen

et al., 2018). Nevertheless, most studies focus on methane production by anaerobic methanogenic archaea. Archaeal
methanogens belong to the phylum *Euryarchaeota* and are distributed over seven orders (Dean et al., 2018). However,
additionally the phyla *Halobacterota*, *Thermoplasmatota* and *Bathyarchaota* are recently discussed as potential methanogens,
especially in peat soil (Bräuer et al., 2020). Methane consumption mitigates the release of methane and was historically thought
to be limited to aerobic bacteria (Söhngen, 1906; Whittenbury et al., 1970), belonging mainly to *Alpha- and*

*Gammaproteobacteria* and *Verrucomicrobia* (Hanson and Hanson, 1996; Op den Camp et al. 2009). However, the so-called
methanotrophs can also be archaea that inhabit anaerobic zones (Boetius et al., 2000; Conrad, 2009; Nazaries et al., 2013;
Dean et al., 2018). In the presence of sulfate and at low $H_2$ concentrations, certain anaerobic methanotrophic archaea (ANME)
can reverse methanogenesis in close interaction with SRB with whom they form symbiotic consortia (Hoehler et al., 1994;
Hansen et al., 1998; Boetius et al., 2000). Both partners benefit from the transfer of intermediates, such as $CH_4$ as electron

donor and carbon source for sulfate reduction, and sulfate as EA for methane oxidation (Hansen et al., 1998). Besides sulfate,
other EAs such as nitrate or metal oxides can play a role in anaerobic oxidation of methane (AOM), especially in coastal
freshwater and brackish wetlands (Segarra et al., 2013), which, unlike ombrotrophic bogs, are not generally poor in alternative
EA (Damman, 1978; Dettling et al., 2006). Sulfate-independent AOM was, for example, reported from freshwater wetlands
(Segarra et al., 2015). Besides archaea, also bacteria like the recently cultured *Candidatus Methylomirabilis oxyfera* from the

NC10 phylum is able to oxidize methane anaerobically using nitrite as an alternative EA (Ettwig et al., 2010), and other genera
within the order Methylomirabilales may also be able to perform this process (He et al., 2016).
Whilst sulfate-mediated $CH_4$ suppression effects are well known in natural coastal wetlands, these mechanisms can be
suspended by the land-use history of degraded coastal peatlands: Koebsch et al. (2019) found that sulfate was depleted in the
coastal fen (Hütelmoor) we are investigating here, except for some local relicts at peat layers below 30 cm depth. These locally

high pore water sulfate concentrations could however not prevent high $CH_4$ emissions from the same fen (Glatzel et al., 2011;
Hahn et al., 2015; Koebsch et al., 2015). Jurasinski et al. (2018) concluded that unlike in marine systems, spatial separation of
methanogenesis and sulfate reduction can sustain methane production and prevent anaerobic methane oxidation in rewetted
coastal fens. This is because methane is formed above the sulfate reducing zone due to a freshening of the surface waters. Like
drought-induced salinization (Kinney et al., 2014; Chamberlain et al., 2020), the inflow of sulfate-containing brackish water

could increase the availability of sulfate and, thus, lead to lower $CH_4$ emissions. The brackish water inflow into the Hütelmoor





in 2019 was followed by a 87 % reduction in CH$_4$ emissions compared to the reference period 2014-2017, while a preceding drought in 2018 lead to a drop in CH$_4$ emissions of 22% (Koebsch et al., 2020).

While CH$_4$ emissions in rewetted freshwater peatlands have been widely studied, the effect of brackish water inflow events on the methane-cycling community and the related biogeochemical patterns in the soil are largely unknown. Field studies of 105 coastal peatlands that cover the transition from freshwater to brackish state are still sparse, and to our knowledge, no study examined the integrated effect of brackish water inflow on biogeochemistry, microbiology and methane emissions, so far.

We thus investigated how microbial communities and coupled peat biogeochemistry change in a freshwater rewetted coastal fen after a brackish water inflow and how this relates to local methane fluxes. Since our study site was exposed to a severe drought in 2018, we put our results in context with potential legacy effects of the preceding drought. We hypothesized that the 110 brackish water inflow will have replenished the sulfate reservoir in peat soil regions relevant for methane production and oxidation. Further, we expected the abundances of sulfate reducing bacteria to increase at the expense of methanogens after the inflow of brackish water. This, in conjunction with an anticipated increasing abundance of sulfate-dependent anaerobic methanotrophic archaea (ANMEs) should decrease methane production and, therefore can explain the reported decrease in methane emissions.

## 2 Material and Methods

### 2.1 Site description

The study site and nature reserve "Heiligensee und Hütelmoor" is located near the city of Rostock at the German Baltic Sea coast. Mean annual temperature at the study area (hereafter "Hütelmoor") was 9.6°C and mean annual rainfall was 635 mm (1991 - 2020, derived from the freely available grid product of the German Weather Service (DWD), for which 1km gridded 120 data are extrapolated from weather station data according to Müller-Westermeier, 1995)). The Hütelmoor is a minerotrophic coastal paludification fen that was drained and used for agriculture between the 1970s and 1990s (Koch et al., 2014; Hahn et al., 2015; Unger et al., 2021). Drainage led to water tables up to 1.60 m below surface (Glatzel et al., 2011) and to rapid peat decomposition (Koch et al., 2017). Therefore, the peat soil in the Hütelmoor is highly degraded (Voigtländer et al., 1996; Hahn et al., 2015) and peat thickness varies between 0.2 and 3 m (Wen et al., 2018; Koebsch et al., 2020). Active drainage of the 125 area by pumping ended in 1990 and resulted in a rise of the water table to 0.3 m below ground (Glatzel et al., 2011), mostly due to freshwater from rising groundwater levels (Miegel et al., 2016). However, effective rewetting with permanent water levels above the ground surface was only achieved after installing a groundsill at the outflow of the catchment in 2009 (Miegel et al., 2016). Emergent macrophytes like *Phragmites australis*, *Carex acutiformis*, *Bolboschoenus maritimus* and *Schoenoplectus tabernaemontani* soon dominated the vegetation, while some large areas of open water remained (Koch et al.,



2017). In 2018, a severe drought caused the water table to decrease and new species like *Tephroseris palustris* and *Ranunculus scleratus* that before had only minor cover in the area colonized the bare peat patches (Koebsch et al., 2020).

In addition to active drainage, a coastal protection dune built in 1963 (Voigtländer et al., 1996; Koebsch et al., 2013) reduced the input of brackish water. The last major brackish water inflow before 2019 occurred in 1995 (Bohne and Bohne, 2008). In 2000, maintenance of the coastal protection dune was discontinued to reinstate the natural flooding regime, leading to a slow

decline in dune height and extent over the years. A storm surge destroyed parts of the rests of the former coastal protection dune close to the lake "Heiligensee" in January 2019, resulting in brackish water inflow into the peatland that potentially changed the formerly only weak pore water salinity gradient (Koebsch et al., 2019) along a previously sampled transect from HC1 towards HC4 (Fig. 2).

## 2.2. Field data collection

We combined data from previous studies with own data recording at the site to evaluate the effect of the inflow of Baltic Sea waters during the storm surge in January 2019 (Fig. 2). We conducted fieldwork at the same four locations (HC1, HC2, HC3 and HC4) discussed in Koebsch et al. (2019) and Wen et al. (2018), which cover different salinity regimes, especially in deeper layers of the peat. Details regarding field sampling protocols (peat biogeochemistry and microorganisms) and data analysis can be found in Wen et al. (2018) and Unger et al. (2021).


In our study, previous data from 2014 served as a baseline and represent the conditions in the freshwater rewetted fen. Therefore, we refer to the geochemical, trace gas and microbial data from Wen et al. (2018) as "Baseline2014" (and "base14" in the figures). Data from Unger et al. (2021) provided insights into the dynamics during the drought in 2018 at location HC2 and are referred to as "Drought2018" (and "drought18" in the figures).

In order to track the surface flow and exchange processes in the above ground water column after the inflow, we complemented the pore water sampling with surface water measurements. Surface water EC measurements took place on several days directly after the inflow in January 2019 to cover the immediate effect of the inflow. On April, 16th 2019, surface water *in-situ* variables and samples and local GHG flux measurements had been taken at the four locations. Sampling for microbial as well as for pore and surface water analysis combined with GHG measurements took place on November 28th and December 2nd, 2019,

hereafter referred to as "Post-inflow Autumn2019". Soil cores and pore water samples were also taken on May 16th, 2019 ("Post-inflow Spring2019") for better comparison with the previous drought study (Unger et al., 2021). This sampling was, however, only done at one of our sampling locations (HC2, see Fig. 2). We derived groundwater level data from a data logger and pressure transducer (Dipper PTEC, SEBA, Kaufbeuren, Germany) installed permanently near location HC2 at 0.49 m depth. Measurements were recorded every 15 min since January 2018.


At each of the four locations, we collected surface and pore water samples for sulfate and chloride concentration analysis. Surface water samples were filtered directly in the field and stored at -25 °C in the lab until further analysis. We used filters





with a pore size of 0.45 µm (Sarstedt, Nümbrecht, Germany) in order to include only dissolved organic carbon (Thurman, 1985; Fiedler et al., 2008). For reference IC measurements, used to compare with data measured during drought in 2018

(Ibenthal, 2020), we filtered 10 mL samples *in situ* through a 0.20 µm cellulose acetate membrane. Surface water electrical conductivity (EC) and pH-values were measured *in-situ* (ProDSS, YSI, Ohio, USA). For pore water sampling in May 2019, pre-filled diffusion pore water samplers were used (Höpner, 1981). We installed pore water samplers well in advance, on March, 28[th] 2019, to allow time for equilibration with the surrounding soil. For the pre-fill to match the salinity of the pore water, we mixed filtered (CA 0,45µm, GE Healthcare Life Sciences Whatman TM, Vancouver, Canada) tap water (which is

river filtrate from river Warnow in Rostock) with filtered Baltic Sea water until salinity of Hütelmoor surface waters was obtained. The diffusion samplers were pre-filled under argon atmosphere and wrapped until installation in the field. Pore water sampling in November and December of 2019 was carried out with peat soil cores, taken in plastic liners (length: 60 cm; inner diameter: 10 cm). Afterwards pore water extraction was conducted using rhizon® pore water suction samplers (Rhizosphere Research Products, Wageningen, The Netherlands, 0.12µm pore size; Seeberg-Elverfeldt et al., 2005). In pre-drilled holes of

the plastic liner, rhizons were inserted and attached to 10 ml syringes. Values of pH and salinity of pore waters were measured immediately after recovery using a hand-held pH-meter (Handylab pH11, Schott Instruments GmbH, Mainz, Germany, calibrated with Mettler Toledo Buffer solutions) and a refractometer (MASTER-S/Millα, Atago, Master-S Refraktometer, Tokio, Japan).

Local CH$_4$ flux measurements were conducted manually using an opaque floating chamber and a portable laser-based analyzer (Picarro G4301, GasScouter, Santa Clara, USA). The floating chamber was 22 cm high and had a total volume of 9953.3 cm$^3$. Flux measurements lasted between 180 and 300 seconds and were repeated three times at randomly chosen spots on the open water body, close to each sampling location. In parallel, we measured chamber and soil temperatures, surface water level, and relative air humidity.


At each location, two peat cores were taken with a Russian D-corer (De Vleeschouwer et al., 2010) and divided into the following depth sections: 5-20, 20-40 and 40-50 cm. Each peat core was used to extract samples for microbial and pore water analysis *in situ*. From each core section we took sediment plugs for peat soil GHG concentration measurements using a tip-cut syringe (Omnifix, Braun, Bad Arolsen, Germany) to get a distinct sediment volume of 3 ml. We immediately inserted the

sediment plugs into 20 ml glass vials (Agilent Technologies, 5182-0837, Santa Clara, USA) completely filled with saturated NaCl for conservation (no head space). We closed the vials air-tight with rubber stoppers and aluminum crimpers, and stored the samples upside-down to avoid gas escape. Per location and core section, we extracted an additional 1 ml soil sample with a 5 ml tip-cut syringe (Omnifix, Braun, Bad Arolsen, Germany) to be analyzed for bulk density to obtain estimates for peat porosity. To prevent drying, the syringe opening was covered with parafilm® (Bemis, Neenah, WI, USA) and samples were

cooled at approximately 4 °C until further analysis. For microbial analysis, we collected subsamples from all core sections mentioned above and additionally the surface layer between 0-5 cm. We placed them into centrifugation tubes (25ml, Falcon®,





Corning Inc, Tewskury, MA, USA) using sterile equipment. We assured immediate cooling on ice and further storage at -80 °C to preserve total nucleic acids until further analyses.

**2.3 Lab analyses: Water and peat**

**2.3.1 Peat soil greenhouse gas concentrations**

We measured peat soil $CH_4$ and $CO_2/H_2CO_3$ (in the following for simplicity referred to as $CO_2$) concentrations using a gas chromatograph (Agilent Technologies 7890A, Santa Clara, USA). A headspace of 3 ml filled with helium was created in the glass vials containing the sediment plugs and put onto a shaker for at least 24h. With a needled syringe we extracted 300 µl of

the headspace volume and inserted 250 µl into the GC using a FID for $CH_4$ and a TCD for $CO_2$ concentration measurement. Gas partial pressures as obtained from the headspace analyses were converted to micromolar concentrations of dissolved $CH_4$ and DIC using the following Eq. (1):

$$\left(\frac{G*H}{T*R*V*P}\right) * 1000 ,$$ (1)

where G is the headspace gas mole fraction (ppm), H is the headspace volume (3 ml), T is the absolute temperature (295.15

K), R is the universal gas constant (0.0821 $L*atm*K^{-1}*mol^{-1}$), V is the peat volume (3 ml) and P is the peat porosity (ml $cm^3$).

**2.3.2 Isotopic composition of dissolved methane and inorganic carbon**

The isotopic composition in the C gases can help to uncover the sources and/or production pathways. We determined $\delta^{13}C$ in $CH_4$ and total $CO_2$ (DIC) after acidification with appropriate volumes of 2M HCl to pH < 4.5 in diluted headspace samples from the glass vials described above (final volume of ~20 ml), using cavity ring-down spectroscopy (CRDS; analyzer model

Picarro G 2201-i) and the Small Sample Isotope Module (SSIM; both Picarro Instruments, Sunnyvale, USA). To exclude spectral interference with hydrogen sulfide potentially present in the samples, we added 1 ml of a saturated Zn-acetate solution (Zn-acetate dihydrate, >98 %; Sigma Aldrich, Taufkirchen, Germany) to fix hydrogen sulfide as solid ZnS. We used the data on the concentrations of gases in the headspace (see above Sect. 2.3.1 to determine a suitable dilution in synthetic air for isotope measurements to fall into the measurement range of the instrument of 300-2000 ppm for $CO_2$ and 2.5-2000 ppm for

$CH_4$. While a maximum of 5 ml of headspace sample could be retrieved and an injected volume of 15-20 ml was necessary, the isotopic composition could not be determined for samples containing less than 10 ppm $CH_4$ in the headspace. Isotope values are expressed in the common δ-notation vs. V-PDB. The values given in per mill (‰) are equivalent to 'mUr' (milli urey; Brand and Coplen, 2012). Calibration for $^{13}C$ in $CH_4$ was done using a working standard of 1000 ppm $CH_4$ (-42.48 ‰) and four certified standards of 2500 ppm $CH_4$ (-38.30, -54.45, -66.50 and -69.00 ‰). For $CO_2$, a working standard of 1000

ppm (-31.07 ‰) and dilutions of pure $CO_2$ (-27.10 and -4.55 ‰) were used. All gas standards if not with certificate had been calibrated against reference materials from IAEA (RM8562) using elemental analysis coupled to isotope ratio mass





spectrometry (EA 3000, Eurovector, Redavalle, Italy; Horizon, NU Instruments, Wrexham, UK). Certified standards were obtained from Air Gas (Air Liquide, Plumsteadville, PA, USA) or from Isometric Instruments (GASCo, Victoria, BC, Canada).

### 2.3.3 Ion composition in pore and surface waters

Sulfate concentrations in pore waters were analyzed by ICP-OES (ICP-iCap 7400 Duo MFC ICP Spectrometer, Thermofisher Scientific, Dreieich, Germany) with a matrix matched external calibration (diluted Atlantic sea water from OSIL (www.osil.co.uk)), and Sc as an internal standard. Precision and accuracy were checked with spiked SLEW-3 (National Research Council Canada Measurement Science and Standards, Ottawa, Canada) and were better than 4.7 and 7.6 % (von Ahn et al., 2021), respectively. Dissolved sulfide was measured in the solutions preserved with Zn acetate on-site following the

methylene blue method of Cline (1969) using a spectrophotometer (SPECORD 40, Analytik Jena GmbH, Jena, Germany). For sulfate and chloride in surface water samples the same method was applied as for the pore water sulfate analysis. Reference surface water sulfate and chloride concentrations, used for comparison with surface water data from drought 2018 (Ibenthal, 2020), were determined with anion chromatography (DX320, Dionex) with inline dilution and dialysis setup (Metrohm 930 Compact IC Flex with a Metrosep A Supp 5-150/4.0 (6.1006.520) column, Herisau, Switzerland).

### 2.3.4 Peat physical properties

The sampled 1 ml soil cores were pushed out of the syringes and weighed after drying for 24 hours at 70 °C to determine bulk density $\rho_b$. Loss on ignition (LOI, in %) was determined for each sampling site and depth section on additional cores at 550 °C using a CEM Phoenix Black Microwave Muffle Furnace (North Carolina, USA). Porosity $\phi$ was then calculated with Eq. (2), following DIN 19683-14 (2007):

$$\phi = 1 - \frac{\rho_b * 100}{\rho_{s-org} * LOI + \rho_{s-min} * (100 - LOI)} \tag{2}$$

with the particle density of the organic material $\rho_{s-org} = 1.40$ g/cm³, and that of the ignition residue $\rho_{s-min} = 2.65$ g/cm³.

### 2.4 Lab analyses: microbial data

### 2.4.1 DNA and RNA extraction

We extracted DNA from 150–200 mg soil from biological duplicates per sampling location and depth section according to the

manufacturer's protocol (GeneMATRIX Soil DNA Purification Kit, Roboklon, Berlin, Germany). DNA concentrations were quantified using a Qubit 2.0 Fluorometer (ThermoFisher Scientific, Darmstadt, Germany), following the protocol of the DNA High Sensitive and Broad Range Assay Kit (ThermoFisher, Berlin, Germany).

For RNA extraction, we required 2 g of soil and used the RNeasy PowerSoil Total RNA Kit (Qiagen, Venlo, Netherlands).

RNA concentrations were also quantified with the Qubit 2.0 Fluorometer and the RNA HS Assay Kit. To remove unwanted excess DNA from RNA samples we used the TURBO DNA-free Kit (Invitrogen, ThermoFisher, Berlin, Germany) according



to in-house protocol. Here, we applied 0.1 % volume (e.g. 5 µl) of 10xTurboDNase Buffer and 1 µl TurboDNase to the extracted RNA dissolved in 50 µl RNase and DNase free water. After mixing, the solution was incubated at 37 °C for 30 min. DNase Inactivation Reagent (5 µl) was added and mixed well using a vortex. After incubation at room temperature for 5 min, 260 the resulting suspension was centrifuged at full speed (17000 g) at 4 °C for 1.5 min. RNA was dissolved in supernatant and separated from the pelleted DNA. RNA concentrations were quantified using Agilent 4150 Tapestation system and RNA Screentape Assay (Agilent, Santa Clara, USA) according to manufacturer's protocol.

### 2.4.2 cDNA synthesis

cDNA synthesis was done using SuperScript III Reverse Transcriptase (Invitrogen, Thermofisher, Berlin, Germany). We 265 followed the in-house protocol and applied 1 µl Random Hexamer and 1 µl 10mM dNTP Mix (nucleotides) onto 50 ng RNA template and filled the tube to a final volume of 13 µl with sterile water. We heated the mixture at 65 °C for 5 min and immediately chilled on ice afterwards. Then, we added 4 µl 5xFirst Strand Buffer, 1µl M DTT, 1 µl Sterile Water and 1 µl SuperScript III RT and mixed well. The resultant mixture was incubated at 25 °C for 5 min following incubation at 50 °C for 60 min. The reaction was inactivated by heating to 70 °C for 15 min.

### 270  2.4.3 PCR amplification and sequencing

Amplification via polymerase chain reaction (PCR) of 16S rRNA genes of DNA and cDNA samples was performed using the universal primer combination Uni515-F/ Uni806-R for both, bacteria and archaea, and primer combination S-D-Arch-0349-a-S-17/ S-D-Arch-0786-a-A-20 for precise archaea detection. For the PCR (Thermal Cycler, T100, Biorad, Feldkirchen, Germany) we added PCR-Buffer, 1.25 U OpitTaq DNA Polymerase, 0.2 mM dNTP, 0.5 mM $MgCl_2$ and 0.5 mM of each 275 primer to sterile water and 5 µl purified sample. The PCR program for universal primers included initial denaturation at 95°C for 5 min and then 30 cycles of denaturation at 95 °C for 30 sec, annealing at 56 °C for 30 sec and elongation at 72 °C for 1 min, followed by a final elongation at 72 °C for 7 min. The PCR program for archaea primer included 35 cycles of denaturation at 95 °C for 30 sec, annealing at 55 °C for 30 sec, elongation at 72 °C for 1 min. Initial denaturation and final elongation were the same as mentioned above. When we could not detect a clean product, we increased the number of PCR cycles to up to 10 280 additional cycles for archaea primer samples. The same PCR program was run on purified RNA extracts to exclude remnants of DNA.

The PCR products were cleaned using Agencourt AMPure XP magnetic bead solution (Beckman Coulter, Massachusetts, USA) according to manufacturer's protocol. Identification of single samples was possible due to unique barcodes, which were attached to the primers. Illumina MiSeq Sequencing was done by Eurofins Genomics (Ebersberg, Germany) with 300bp paired-285 end mode.



### 2.4.4 qPCR gene abundance measurements

To quantify the abundances of the target genes 16S rRNA, *mcrA*, *pmoA* and *dsrB*, we used quantitative PCR (qPCR, CFX Connect Real-Time PCR Detection System, Bio-Rad, München, Germany) with the double-strand binding dye SYBR Green (KAPA universal). Whereas primers for 16S rRNA (Eub341-F/Eub534-R) target general prokaryotic microorganisms, primers

used to amplify *mcrA*, *pmoA*, and *drsB* are specific for enzymes of methanogenic archaea (*mcrA,* mlas-F/mcrA-R), aerobic methanotrophic bacteria (*pmoA,* pmoA189-F/pmoA661-R) and sulfate reducing bacteria (*dsrB,* DsrB2060-F/DsrB4-R). According to the in-house protocol, we used 10 µl of SYBR Green, 0.08 µl of each Primer, 5.84 µl sterile water and 4 µl template per reaction. The qPCR program included initial denaturation at 95 °C of 3 min, denaturation at 95 °C for 3 sec, annealing for 20 sec, elongation at 72 °C for 30 sec and a plate read at 80 °C for 3 sec to create the melting curve. Annealing

temperatures were 60°C for 16S rRNA, *mcrA* and *pmoA* and 62°C for dsrB, respectively. Standard curve was typically based on a series of dilutions e.g. from $10^8$ to $10^3$ (Wen et al., 2018; Unger et al., 2021). We performed 35 qPCR cycles for 16S rRNA and 40 cycles for *mcrA*, *pmoA* and *dsrB*. Since Wen et al. (2018) and Unger et al. (2021) did not investigated sulfate reducing bacteria, we performed qPCR with *dsrB* target primers additionally with material from Baseline2014 and Drought2018 study.

## 2.5 Data analysis

### 2.5.1 GHG flux estimation

Flux data analyses were done in R (R Core Team 2021). CH4 fluxes were estimated with function fluxx of the R package 'flux' (Jurasinski et al., 2014) as described in Huth et al. (2012), Günter et al. (2017) and Huth et al. (2021). We used the atmospheric sign convention, meaning that positive fluxes indicate a release from the ecosystem to the atmosphere and negative

fluxes indicate uptake by the ecosystem.

### 2.5.2 Processing of microbial sequence data

The Illumina paired-end (PE) sequences were preprocessed by the method described in Krauze et al. (2021) and Yang et al. (2021). Briefly, demultiplexing was implemented by combining mothur (version 1.39.0) (Schloss et al., 2009), BBTools (Bushnell, 2014) and a custom python script. The PE reads were processed with the 'make.contigs' function of mothur and the

resultant report and groups files were parsed with a custom python script to get sequence identifiers of the good quality contigs (minimum overlap length > 25, mismatch bases <5 and without ambiguous base) for each sample. Next, PE sequences were extracted for each sample with the 'filterbyname.sh' function of BBTools. After these steps, orientation of PE sequences was corrected by 'extract_barcodes.py' function of QIIME (version 1.8) (Caporaso et al., 2010). After removing primers with awk command, the PE sequences were fed to DADA2 (Callahan et al., 2016) for filtering, dereplication, chimera check, sequence

merge, and amplicon sequence variants (ASV) calling. Taxonomic assignment was referred to SILVA138 (Quast et al., 2013) in platform QIIME2 (Bolyen et al., 2019).



### 2.5.3 Visualization and statistical analyses

Several R packages (simba, Jurasinski and Retzer, 2012), dplyr (Wickham et al., 2021), reshape2 (Wickham, 2007), forcats
(Wickham, 2021), scales (Wickham and Seidel, 2020)) were used to visualize and explore microbial community data in R. We
followed Unger et al. (2021) and used bubble plots and NMDS ordinations to investigate changes in the microbial community
structure and in the structure of the communities of specific groups (methanogens, methanotrophs, SRB and ANME) of
different sites and depths with the DNA and cDNA data. We provide the R workflow as supplemental material. In brief, we
constructed bubble plots to visualize dissimilarity in microbial community composition at the order (methanogens,
methanotrophs), class (SRB) and genus (ANMEs) level among subsites and across sampling periods. We extracted the relevant
groups from the data by searching for text strings using regular expressions across the whole taxonomy for the DNA, and
where available, for the cDNA data. Then we transformed the data from wide format to molten format and plotted the bubble
plots using ggplot2 (Wickham, 2016) and arranged them using ggpubr (Kassambara, 2020). To account for the different sizes
of the bacterial and the archaea datasets and strongly varying count numbers across taxonomical units, we used Wisconsin
double standardization at each group level (vegan package, Oksanen et al., 2020).
Further, we built NMDS plots by using the function *metaMDS()* of R package vegan (Oksanen et al., 2020) at the domain level
to examine differences in relative abundances of bacteria and archaea over sampling locations and time (sampling campaigns).
Here, we applied Wisconsin double standardization on the entire bacterial and archaeal dataset each before running the NMDS.
Colors were used according to colorblind-friendly palette from ggthemes (Arnold, 2021).
To visualize quantitative differences in functional gene abundances, we created depth profiles using the R packages ggplot2
(Wickham, 2016), tidyr (Wickham, 2021), ggpubr (Kassambara, 2020) and patchwork (Pedersen, 2020). In addition, we
created depth profiles of pore water variables such as pH, electrical conductivity, dissolved gas, sulfate and chloride
concentrations and gas isotopic signatures. To test for differences in average values, we used ANOVA and a post-hoc Tukey
Test for normally distributed data. For not normally distributed variables, we used the Kruskal-Test and Wilcox-Test (including
bonferroni correction) as a post-hoc test for more than three subgroups and Mann-Whitney-U-Test for exactly two variables.
To display average values for different subgroups we used psych package (Revelle, 2020).

## 3 Results

### 3.1 Brackish water effect on surface and pore water geochemistry

The year 2019 after the brackish water inflow in January had a mean annual air temperature of 10.7 °C and an average annual
precipitation of 605 mm. Thus, 2019 was warmer (+ 1.1 °C) and slightly drier (-30 mm) than the averages of the latest 30 years
reference period (1991-2020, DWD Germany, see Sect. 2.1). Since rewetting in 2009 water levels resided largely above ground
surface year-round. During the drought in 2018, however, mean water levels near station HC2 ranged between -0.61 (below



ground) and 0.59 m (above ground) and stayed below ground surface for 153 days. In 2019, water levels showed much less variation and ranged between -0.07 and 0.36 m, dropping to below ground surface on only 2 days.


The inflow event in January 2019 created a pronounced lateral brackish zonation in the surface water, which was essentially shaped by the separating effect of the main ditch, which crosses the area in NE-SW direction: HC3 and HC4, located north of the ditch and closest to the Baltic Sea, had highest electrical conductivities (EC) > 22 mS/cm, whilst HC1, located south of the ditch and furthest inland, had lowest EC values (≤ 11 mS/cm, Fig. 2). The data measured during our Post-inflow Spring2019

campaign in April/May at the distinct transect stations were in line with these initial inflow patterns, although the EC had decreased significantly overall since January. Surface water EC decreased down to 12mS/cm in vicinity to the Baltic Sea at HC3 and HC4 and down to 7 mS/cm at the inland spots HC1 and HC2. By autumn (Post-inflow Autmn2019), EC values at HC3 and HC4 dropped down to 8.7 mS/cm, while the EC values at the inland locations HC1 and HC2 were 5.3 and 6.3 mS/cm, respectively. As a result, the lateral brackish water gradient that had established due to the inflow event (Fig. 2) had largely

levelled out within eleven months. In parallel to EC, surface water sulfate concentrations also decreased from 3.9 mM (HC4) and 2.7 mM (HC1) down to 1.2 mM and 0.2 mM, respectively between Post-inflow Spring2019 and Autumn2019. Unlike sulfate concentrations, which decreased throughout all locations from Post-inflow Spring2019 to Autumn2019, chloride concentrations only decreased at the inland locations HC1 and HC2 and increased at HC3 and HC4, the locations closer to the Baltic Sea (Table S1). This created a lateral span from HC4 (47.3 mM) towards HC1 (12.4 mM) in Post-inflow Autumn2019,

which did not occur in Post-inflow Spring2019. The divergent temporal dynamics of surface water chloride concentrations at different areas of the peatland, were also reflected by the sulfate/chloride ($SO_4^{2-}$/ $Cl^-$) ratios: $SO_4^{2-}$/ $Cl^-$ ratios at all locations in Post-inflow Spring2019 were within a narrow range of 0.09-0.12 and decreased towards Post-inflow Autumn2019 (0.01-0.03) by around one order of magnitude and were then highest at HC4 and lowest at HC1 (Table S1).

In the pore water, there was a general and significant increase in EC (from 5.1±2.8 to 9.1±3.3 mS/cm, $p < 0.001$, Wilcox test), sulfate (from 1.1±3.7 to 5.3±6.9 mM, $p < 0.001$, Wilcox test) and chloride concentrations (from 37.8±22.8 to 55.1±22.4 $p <$ 0.05, Wilcox test) after the inflow from Baseline2014 conditions to Post-inflow Autumn2019, averaged over all four locations and across all sampling depths (Fig. 3b, c and d, for average values see Table S1). The individual depth profiles are, however, shaped by their specific location along the lateral brackish gradient and the pre-existing Baseline2014 sulfate concentrations.

Pore water sulfate levels at HC3 and HC4 close to the Baltic Sea increased only moderately from average 0.02 and 0.16 mM at Baseline2014 to 0.8 mM and 0.7 mM in Post-inflow Autumn2019, respectively. In contrast, sulfate concentrations at HC1, furthest inland, increased strongly from a low level of only 0.01 mM (Baseline2014) to 9.9 mM (Post-inflow Autumn2019) and thereby approached the record levels of HC2. HC2 had highest pre-existing Baseline2014 sulfate concentrations at deep pore water layers (Fig. 3g). Averaged across the profile, sulfate concentrations at HC2 increased from 3.5 mM to 8.9 mM after

in inflow (Table S1). In Post-inflow Autumn2019, pore water chloride concentrations increased mostly in upper peat layers at all four locations after the inflow (Fig. 3d), but to a higher extent at locations in proximity to the Baltic Sea (HC3 and HC4





with averages across peat profile: 68.1 and 74.8 mM, respectively) and to a lesser extent at locations HC1 (23.5 mM) and HC2 (57.1 mM). Baseline2014 pore water chloride concentrations showed similar differences in magnitude like sulfate concentrations, but here, concentrations were lower at HC1 (12.4 mM) compared to the other three locations (40.5–47.4 mM, see Table S1).

When zooming in to HC2 (Figure 3e-h), where we have additional sampling data (Drougth2018 and Post-inflow Spring2019), a new sulfate concentration maximum of almost 40 mM becomes apparent in depths below 35 cm during the Drought2018 (Fig. 3g). At the same time, chloride concentrations hardly increased during the drought, but rather in Post-inflow Autumn2019 (Fig. 3h). The additional data at location HC2 also show a gradual increase of EC starting in Post-inflow Spring2019 in the surface layers and increasing throughout the peat profile towards Post-inflow Autumn2019 (Fig. 3f).

### 3.2 Greenhouse gas fluxes, concentrations and isotopic signatures in the pore water

All $CH_4$ fluxes measured in 2019 differed slightly but not significantly among locations despite the differences in surface water EC and sulfate concentrations. Overall, $CH_4$ fluxes averaged (median) 0.06 mg m$^{-2}$ h$^{-1}$ and 0.4 mg m$^{-2}$ h$^{-1}$ in Post-inflow Spring2019 and Post-inflow Autumn2019, respectively (Table S1), and differed significantly between the post-inflow seasons ($p <0.01$, Mann-Whitney-U Test).

Dissolved $CH_4$ concentrations in pore water samples decreased from an average of 232.6±161.8 µM in the Baseline2014 sampling to an average of 158.0±155.4 µM in our Post-inflow Autumn2019 sampling. Whereas $CH_4$ concentrations varied strongly with location and depth in 2014, the depth variation was much lower in Post-inflow Autumn2019, while the variability across locations did not change much (Fig. 3i). Like $CH_4$, also $CO_2$ concentrations decreased from Baseline2014 (9.8±6.9 mM) to Post-inflow Autumn2019 (1.8±1.0 mM) and were significantly different between the two years (Wilcox test, $p< 0.0001$). This strong decrease was associated with a strong decrease in depth-dependent variation (Fig. 3j). At location HC2 (Fig. 3m-n), where additional measurements were taken during the Drought2018 and Post-inflow Spring2019, $CH_4$ concentrations remained relatively high at the surface in Drought2018, but were lower in deeper layers and showed much less depth variation than Baseline2014 (Fig. 3m). Here at HC2, $CH_4$ concentrations decreased from average 297.1±218.6 µM (Baseline2014) to 70.9±114.3 µM (Drought2018) and increased strongly to an average of 325.4±126.7 µM in Post-inflow Spring2019, showing the highest values in almost all sampled depth sections. In Post-inflow Autumn2019, the pore water $CH_4$ concentrations at HC2 decreased again but remained, on average (91.0±68.7 µM), slightly higher than during the Drought2018. Similarly, $CO_2$ concentrations (Fig. 3n) decreased strongly from Baseline2014 (16.6±7.3 mM) to during the Drought2018 (1.1±0.4 mM) in the same pore water samples (HC2), but increased only marginally in Post-inflow Spring2019 (1.4±0.2 mM) and Post-inflow Autumn2019 (2.2±1.5 mM) afterwards. While strong decrease in depth variation of $CO_2$ concentrations were found during the Drought2018 and in Post-inflow Spring2019 shortly after the inflow, depth variation slightly increased in Post-inflow Autumn2019 with highest values with increasing depths (Fig. 3n).

At all locations (Fig. 3k), $\delta^{13}C$-$CH_4$ values were significantly lower (-64.7±4.1 ‰, $p < 0.001$, Tukey test) Post-inflow Autumn2019 compared to Baseline2014 (-60.6±2.6 ‰). $\delta^{13}C$-$CO_2$ values (Fig. 3l) also became more negative after the inflow





and differed significantly between Baseline2014 and Post-inflow Autumn2019, averaging -5.2±5 ‰ and -20.94±2.1 ‰, respectively (p <0.001, Tukey Test). At HC2 (Fig. 3o), $\delta^{13}$C-CH$_4$ decreased steadily from Baseline2014 (max: -57.8‰), Drought2018 and Post-inflow Spring2019 to Post-inflow Autumn2019 (min: -72.6 ‰). It is also apparent that the decrease of $\delta^{13}$C-CO$_2$ took already place during Drought2018 (Fig. 3p), leading to significant differences between the Baseline2014 and the Drought2018 (p < 0.001, Tukey Test) at location HC2. Here, average $\delta^{13}$C-CO$_2$ decreased from -8.4±5.7 ‰ (Baseline2014)

to -19.9±5.1 ‰ (Drought2018), increased in Post-inflow Spring2019 (-15.2±5.2 ‰) and decreased again in Post-inflow Autumn2019 (-21.5±0.9 ‰).

### 3.3 Microbial community composition

Throughout all sampling locations, the most abundant groups of methanogenic archaea belonged to the orders Methanosarciniales, Methanomicrobiales, Methanobacteriales, Methanofastidiosales, Methanocellales and

Methanomassiliicoccales (orange, Fig. 4a). Increases and decreases in DNA-based relative abundances in individual depth sections caused methanogenic archaea to appear more homogenous along the peat profile in Post-inflow Autumn2019 compared to the Baseline2014. While acetoclastic groups, especially Methanosaeta (or Methanothrix (Bräuer et al., 2020) within Methanosarciniales order) remained rather constant, CO$_2$ reducing and some potentially methylotrophic methanogens like Methanomicrobiales and taxa within Methanobacteriales gained in relative abundance after the inflow. Among the

methanotrophic bacteria, the genera Methylocystis and Methylosinus within the order Rhizobiales were most abundant at the Baseline2014 sampling and decreased in Post-inflow Autumn2019, while representatives of the order Methylococcales increased and were subsequently present throughout the whole depth profile from 0-50 cm. *Candidatus* Methylomirabiles oxyfera within Methylomirabilales were found in low abundances at the Baseline2014 sampling and appeared in higher abundances in Post-inflow Autumn2019 at HC1 and HC4 (blue, Fig. 4a).

Sulfate reducing bacteria (SRB) were already present in relatively moderate abundances during the Baseline2014 sampling. The most dominant classes in Baseline2014 were Desulfobacteria and Desulfobaccia (green, Fig. 4a). Other SRB classes such as Desulfovibrionia, Desulfotomaculia and Desulfobulbia were distributed more equally and higher in relative abundance in surface peat soil in Post-inflow Autumn2019 (green, Fig. 4a). The anaerobic methanotrophic *Candidatus* Methanoperedens was found in most locations in high relative abundance (black, Fig. 4a). In Post-inflow Autumn 2019, relative abundances of

*Candidatus* Methanoperedens decreased strongly at location HC3 and HC4. ANME-3 was only present in high relative abundances in very surface peat layer at HC2 in Baseline2014 (black, Fig. 4a). In Post-inflow Autumn2019, ANME-3 was only detected between 20-40 cm at HC3.

At location HC2 (Fig. 4b), a more detailed picture of the microbial communities was possible due to two additional sampling campaigns during the Drought2018 and Post-inflow Spring2019. Despite differences in biogeochemical conditions, it appears

that the microbial community compositions and DNA-based relative abundances at HC2 show similar patterns comparable to the other locations during the Baseline2014 and Post-inflow Autumn2019 samplings (Fig. 4a). According to the data from HC2, most methanogenic orders increased during the drought and remained high in abundance after the brackish water inflow





such as taxa within Methanomicrobiales and Methanobacteriales (orange, Fig. 4b). In contrast, Methanofastidiosales decreased shortly after the inflow (Post-inflow Spring2019) and increased in deeper peat layers towards Post-inflow Autumn2019

(orange, Fig. 4b). Other methanogens, such as Methanomassiliicoccales and Methanocellales, decreased during the Drought2019 (orange, Fig. 4b). Both orders remained low in abundance after the inflow of brackish water. Putatively active methanogen taxa (cDNA-based communities) changed only slightly from the Drought2018 towards the post-brackish water inflow year (Fig. 4c). These changes were mostly in line with the changes observed in the DNA-based data (Fig. 4b).

Within bacterial methanotrophs at location HC2, Methylococcales became more abundant during the Drought2018 and remained high in DNA-based relative abundance as the most dominant methanotrophs after the brackish water inflow (blue, Fig. 4b). In general, the abundances of Rhizobiales decreased during the Drought2018 and even more so in Post-inflow Sping2019 and Autumn2019. Methylomirabilales increased during the Drought2018 in peat layers below 40 cm and disappeared almost completely after establishment of higher water tables post-inflow. The cDNA-based abundances of aerobic

methanotrophic bacteria like Methylococcales were similarly high during Drought2018 and post-inflow conditions while cDNA-based abundances of Rhizobiales were lower (blue, Fig. 4c), which is reflecting the results of DNA analysis (blue, Fig. 4b). Methylomirabilales were not detected in the cDNA-based extractions.

In the DNA-based community profile, most SRBs (Desulfovibrionia, Desulfotomaculia, Desulfobulbia, Desulfobacterota) increased in relative abundance at HC2 during the Drought2018 and remained highly abundant in most peat layers in Post-

inflow Spring2019 and Autumn2019 (green, Fig. 4b). However, classes such as Syntrophobacteria and Desulfobulbia showed higher cDNA-based abundances only after the drought in Post-inflow Spring2019 in the surface peat layers.

DNA-based abundances of *Candidatus* Methanoperedens increased already during the Drought2018 and remained high afterwards through the whole peat profile (black, Fig. 4b). These findings can however not be confirmed with the data on cDNA-based abundances, suggesting no active role of *Candidatus* Methanoperedens except during Drought2018 in the deepest

peat layer at HC2. DNA-based ANME-3, which were detected in surface layers in Baseline2014 at location HC2, were still present during the Drought2018, but with low abundances in Post-inflow Spring2019 and Autumn2019 (black, Fig. 4b). According to the cDNA analysis, active ANME-3 were little abundant in the surface peat layers during the Drought2018 (black, Fig. 4c).

### 3.4 Absolute abundances of microbial groups (qPCR)

Mean total prokaryote gene sequence abundances (16S rRNA) were very similar among the four sampling campaigns, whereas abundance variation along depth sections decreased after the brackish water inflow (Fig. 5a). Absolute methanogenic (*mcrA*) and aerobic methanotrophic (*pmoA*) gene copies (per gram of dried soil) were lower at the surface layer and higher at deeper peat layers in Post-inflow Autumn2019 at all locations compared to Baseline2014 conditions (Fig. 5b and c). Average *mcrA* gene copies did not differ largely across sampling campaigns, but *pmoA* gene copies decreased slightly, but not significantly

towards Post-inflow Autumn2019, despite considerable spatial variability. After the brackish water inflow, absolute *mcrA* gene





abundances of DNA-based analysis were one to two orders of magnitude higher compared to *pmoA* abundances, which is also reflected in the cDNA-based abundances from location HC2 (Table S1). Functional genes encoding for sulfate reducing bacteria (*dsrB*) increased significantly (Wilcox test, p< 0.001) in absolute abundance after the brackish water inflow at all locations (Fig. 5d). Mean *dsrB* gene copy numbers from Post-inflow Autumn2019 were close to three orders of magnitude

higher compared to the Baseline2014.

Zooming in on the higher temporal resolution at HC2 shows that absolute methanogenic abundances (*mcrA*) increased during the Drought2018 and Post-inflow Spring2019 and increased further Post-inflow Autumn2019 (Fig. 5f). Methanotrophs (*pmoA*) also increased already during the Drought2018, but decreased Post-inflow Spring2019 to the level from before the Drought2018 (*pmoA*, Fig. 5g). SRB abundances (*dsrB*) increased slightly during the Drought2018, but increased much stronger

in Post-inflow Spring2019 and especially after additional six months, in Post-inflow Autumn2019 (Fig. 5h).

### 3.5 Microbial community composition similarities through time

The NMDS ordinations (Fig. 6) reveal clustering according to different sampling times and locations. The overall composition of the bacterial communities at different sampling locations and depths show strong similarity across all sampling campaigns (Baseline2014, Drought2018, Post-inflow Spring2019, Post-inflow Autumn2019) as reflected in the substantial overlap of the

polygons in Fig. 6a. Also, the overall archaeal community compositions overlap quite strongly between sampling dates, but a distinct clustering is more clearly visible (Fig. 6b). The bacterial Baseline2014 samples had slightly higher EC and $CO_2$ concentrations and were more enriched in $^{13}C$ $CH_4$ (see post-hoc fit arrow in Fig. 6a) compared to the other sampling campaigns. Those surface samples of HC2 during the Drought2018 with low sulfate concentrations have higher pH values. Although Drought2018 sampling was only conducted at location HC2, the Drought2018 cluster spans a wide range of the

complete bacterial variation and Post-inflow bacterial community composition is almost entirely a subset of it. Post-inflow Spring2019 samples (also only HC2) appeared as a subset of the Post-inflow Autumn2019 samples, when cores were taken at all locations.

Baseline2014 archaeal communities (Fig. 6b) differed from the Drought2018 and Post-inflow (Spring2019 and Autumn2019) clusters and were more variable. At the same time, the $^{13}C$ in $CH_4$ and DIC in the pore water samples was positively correlated

with Baseline2014 samples. Similar to the bacterial Drought2018 communities, also archaeal Drought2018 communities show large similarities with the Post-inflow communities, but do not cover the variations at locations HC1 and HC4 in Post-inflow Autumn2019, where the archaeal communities seem to have been very different from the HC2 communities during the Drought2018. Post-inflow Spring2019 archaeal communities overlapped largely with the Post-inflow Autumn 2019 communities and were a subset like bacterial communities. The communities of the Baseline2014 data and those from the

Drought2018 and Post-inflow 2019 seem to be associated with changes in pore water trace gases, since their isotopic signatures and DIC concentrations were the only physicochemical variables that were significantly correlated with the ordination configuration with a positive change vector in the direction of the Baseline2014 positions. Sulfate-dominated plots are





distinctly clustered within Baseline2014 and Drought2018, but not across sampling campaigns, and seemingly correlate with peat soil depth. Sulfate as a variable was however not significant neither within the bacterial nor the archaeal communities.

## 4 Discussion

### 4.1 Effect of brackish water inflow on surface and pore water geochemistry

The January 2019 storm surge brought brackish water into the freshwater rewetted peatland, but the emerging biogeochemical shift was not equally distributed across the sampled transect. Instead, two zones of different brackish impact, separated by the main ditch, formed with higher EC concentrations close to the Baltic Sea and lower EC concentrations further inland (Fig. 2). Still, large increases in electrical conductivity (EC, 0.6 to 7.6 mS/cm) as well as in sulfate (0.1 mM to 5.6 mM) and chloride (2.9 mM to 55.6 mM) concentrations between April 2018 (Ibenthal, 2020) and April 2019 (this study) were observed in surface water near location HC2. Given that EC, sulfate and chloride concentrations increased at every single location in the pore water from the Baseline2014 sampling to the Post-inflow Autumn2019 sampling (Fig. 3b, c and d), we can assume that the surface water geochemistry also changed at locations HC1, HC3 and HC4 and that all locations were affected by the brackish water inflow in spring 2019 post-inflow despite the different distances to the Baltic Sea. The brackish water inflow is also reflected by sulfate/chloride ($SO_4^{2-}/ Cl^-$) ratios in the surface water that exceeded (0.09-0.12) the ratio of the Southern Baltic Sea coast (0.07, Rheinheimer, 2013) at all locations shortly after the inflow. The fen's surface water $SO_4^{2-}/Cl^-$ ratio decreased from spring towards autumn post-inflow. The reduction was higher at locations HC3 and HC4 than at HC1 and HC2. At location HC1 and HC2, surface sulfate concentration decreased in parallel with chloride concentrations, which might be a result of dilution with freshwater. At HC3 and HC4 close to the Baltic-Sea, surface water EC and sulfate concentrations had decreased between spring and autumn post-inflow while chloride concentrations had not (Table S1). Although we lack direct evidence for increased sulfate reduction rates, we can assume that sulfate was microbially processed in the underlying peat soil indicated by the reduction of surface water sulfate concentrations. Similar to the surface water patterns, pore water chloride concentrations increased at a much higher rate after the inflow in autumn 2019 at HC3 and HC4 compared to HC1 and HC2. However unlike in the surface water, pore water sulfate concentrations also increased post-inflow, but were much lower at HC3 and HC4 compared to HC1 and HC2, suggesting depletion of the sulfate reservoir through microbial sulfate reduction at locations close to the Baltic Sea. This was also seen by the increased absolute abundances of sulfate reducing bacteria (SRB, see Fig. 5d and h).

Despite different lateral patterns among the locations, the shift of pore water biogeochemistry from freshwater in the upper parts to brackish conditions throughout the averaged profiles (Fig. 3b, c and d) was clearly visible by the approximation of upper and deeper pore water sulfate and EC levels after the inflow (Fig. 3b and c). Most probably, new sulfate from the inflow sits on top of the old relicts and will thus help to suppress methane emissions, since sulfate reservoir is no longer depleted (Jurasinski et al., 2018).





The legacy effect of the preceding drought should nevertheless be accounted for its influence on the changes in sulfate
concentrations. At location HC2, pore water sulfate concentrations were already higher than at the other three locations, where
sulfate was almost completely exhausted during our baseline study in 2014 (Koebsch et al., 2019). During the drought year
2018, sulfate concentrations at HC2 increased further up to 40mM, which was most likely due to the oxidation of existing
sulfides yielding higher sulfate concentrations than originally present in the pore water (Boman et al., 2008; Boman et al.,
2010). When the drought-induced drop of the water levels recovered parallel to the brackish water inflow, sulfate levels at
HC2 decreased (Fig. 3g). Most likely, the water table reservoir got filled up with freshwater before the inflow, reducing the
effect that the inflow of sulfate-containing brackish water may have had on the sulfate concentrations in the pore water. Still,
sulfate concentrations remained higher than the baseline 2014 levels observed at all locations (Fig. 3c), including the remote
location HC1 most distant from the Baltic Sea. Further, chloride as a conservative tracer also increased after the inflow at all
four locations (Fig. 3d), which is most unlikely due to drought-induced salinization. Therefore, the drought cannot be the only
source for the observed increase in pore water ion concentrations and hence, we can assume that both, brackish water inflow
and not only the legacy effect of the drought in 2018 changed sulfate concentrations in the surface and pore water and was
critical for the methane dynamics and the microbial community composition.

## 4.2 Effect of brackish water inflow on greenhouse gas pools in the peat soil

The $\delta^{13}$C-CH$_4$ values decreased after brackish water inflow (in autumn 2019, Fig. 3k) indicating isotopically lighter, newly
produced CH$_4$ and suggesting that the formation of $^{13}$C depleted methane increased slightly, potentially as a result of shifts in
methanogenic pathways. Microorganisms preferentially take up isotopically lighter substrates ($^{12}$C) and leave heavier
substrates ($^{13}$C) in the soil, so values that are more negative indicate more microbially-produced $^{12}$C-CH$_4$ e.g. during
methanogenesis (Oremland, 1988). At the same time, the overall decrease in CH$_4$ concentration post-inflow (Fig. 3i) suggests
less methanogenesis after the inflow, assuming no major changes in effluxes. This is despite the observed increase of CH$_4$
concentrations only five months after the inflow at location HC2, but not in autumn 2019 (Fig. 3m). CH$_4$ concentrations in the
peat soil decreased during the drought (Unger et al., 2021), increased shortly after the brackish water inflow and decreased
again after another 7 months (Fig. 3m). During the drought, aerobic conditions and re-oxidation of terminal electron acceptors
likely hampered methanogenesis (Achtnich et al., 1995; Dettling et al., 2006). However, methane production must have been
triggered again at the beginning of 2019 due to the water table increase, availability of substrates (Koebsch et al., 2020), and
the re-establishment of anaerobic conditions (Whiting and Chanton, 1993; Popp et al., 1999). In autumn 2019, methanogens
likely became substrate- and temperature-limited, and thus, CH$_4$ concentrations might have decreased for these reasons.
We do not observe indicators for sulfate-driven anaerobic CH$_4$ oxidation based on the isotopic signatures and microbial
community data (see below) after the inflow as anticipated. Specifically, we expected a clear drop in peat methane
concentrations and a shift towards more positive values, because microbes also take up lighter $^{12}$C-CH$_4$ for methane oxidation
rather than the heavier $^{13}$C-CH$_4$ (e.g. Whiticar et al., 1986; Oremland, 1988; Meister et al., 2019). Instead, we see a shift
towards more negative $\delta^{13}$C-CH$_4$ values (Fig. 3k). While this seemed contradictory at first sight, this may be explained by the





observation that methane produced under thermodynamically more unfavorable conditions, e.g. in microenvironments (Knorr et al., 2008), tends to be more depleted in $^{13}$C (Penning et al., 2005).

The lower $\delta^{13}$C-DIC between the baseline sampling in 2014 and autumn 2019 sampling post-inflow indicate an increase in
non-methanogenic $CO_2$ production (Fig. 3l). The isotopic signatures from HC2 suggests that a depletion of $^{13}$C in the DIC pool already took place during the drought year 2018 (Fig. 3p). Since aerobic conditions fuel decomposition, and enhance the diffusivity of the peat soil, $CO_2$ production might have therefore increased (Alm et al., 1999). In spring 2019, shortly after the brackish water inflow, DIC became less depleted in $^{13}$C (potential onset of methanogenesis, Fig. 3p) compared to drought conditions, especially in the peat layer below 20 cm. However, in autumn post-inflow, values of $\delta^{13}$C-DIC decreased down to
the level of the drought (Fig. 3p), indicating increased $CO_2$ production. This can be attributed to the increase of the water table and potentially non-methanogenic $CO_2$ production (Knorr et al., 2008). Other than via aerobic peat decomposition during the drought or via methane oxidation, $CO_2$ may be produced more intensively via sulfate reduction after the inflow of the sulfate-rich brackish water. In addition, $\delta^{13}$C-DIC values from autumn 2019 approached -27 ‰ (Fig. 3l and p), which is close to the average values of the most dominant plant species (C3 plants, Meyers, 1994) indicating non-methanogenic pathways of $CO_2$
production (Boehme et al., 1996; Corbett et al., 2013).

In contrast, concentration measurements in the pore water (Fig. 3n) show that $CO_2$ levels decreased with the drought compared to the baseline sampling, and remained low after the inflow. Unlike the isotopic signatures, this indicates that $CO_2$ production was rather low during the drought and after the inflow. However, trends in trace gas concentrations and isotopic signatures can also appear contradicting, because gas concentrations are temporally highly variable and might not reflect biogeochemical
processes, since downstream processes likely use up intermediate products or gases get emitted to the atmosphere. This means that $CO_2$ production might be higher during drought and further on, but the produced $CO_2$ might not accumulate and is therefore not measureable. Increases in $CO_2$ emissions from ecosystem respiration during the drought support this hypothesis (Koebsch et al., 2020).

Overall, the strong depletion of $^{13}$C in $CH_4$ and the slight decrease in concentrations indicates that methanogenesis did not decrease to an extent that this could explain the measured decrease in $CH_4$ fluxes (Koebsch et al., 2020). Due to persistently high $CH_4$ concentrations, strongly negative isotopic signatures and the patterns in microbial community composition, we can conclude that methane oxidation was of minor importance in the peat soil. Sulfate-mediated anaerobic methane oxidation can also not explain the decrease in pore water isotopic signatures of $\delta^{13}$C-DIC in autumn 2019 post-inflow, indicating higher
anaerobic but non-methanogenic $CO_2$ production, e.g. via sulfate reduction. If anaerobic $CO_2$ production had been a result of methane oxidation, it had to happen in an area outside the scope of our analysis, namely the water column or the fresh litter layer above the peat soil. It is well established that the fresh organic litter in rewetted peatlands can be a hotspot of biochemical cycling (Hahn-Schöfl et al., 2011), providing nutrients (Wang et al., 2015) and shelter for microorganisms (Bani et al., 2018). Therefore, the results indicate a complex impact of drought and subsequent brackish water inflow on the investigated
ecosystem changes with respect to carbon cycling across different spatial compartments.




### 4.3 Effect of brackish water inflow on methane cycling microorganisms

The inflow of sulfate-containing brackish water caused two main changes within the microbial communities: 1) Sulfate reducing bacteria (SRB) increased both in relative and absolute abundance (Fig. 4, 5d and 5h) and 2) methanogenic archaea changed regarding the community composition, but not in absolute abundances (Fig. 4). Changes within the methanotrophic

community after the inflow, both aerobic and anaerobic, were marginal. SRB communities were clearly affected by the inflow of brackish water, because they only increased strongly in absolute abundances after it, but not during the drought (Fig. 5h). This even holds for HC2 where sulfate concentrations were higher during the drought than after the inflow (Fig. 3g).

To determine the direct effect of the inflow on the methanogenic and methanotrophic communities excluding the legacy effect of the drought is more difficult. After the inflow, we observed the highest reduction of methanogenic archaea (*mcrA*) and

aerobic methanotrophic bacteria (*pmoA*) in the upper layers of the peat soil (Fig. 5b and c). In deeper layers, methanogens and bacterial methanotrophs increased in abundance relative to baseline conditions. Zooming in to location HC2, where we measured total and putatively active microbes also during the drought in 2018 and in spring 2019 after the inflow, we must conclude that methanogenic and methanotrophic absolute abundances had changed already during the drought and did not change much further after the brackish water inflow (Fig. 5f and g). Aerobic methanotrophs like Methylococcales were likely

activated under oxic drought conditions (Henckel et al., 2001; Ma et al., 2013; Unger et al., 2021) and remained present, when the water came back after the inflow (Fig. 4b).

The data from HC2 also show that methanogens increased in absolute abundances mostly at peat layers below 5 cm during the drought and remained high afterwards, whereas they decreased in the surface peat (Fig. 5f). At the surface, the reduction of methanogen abundances due to the previous drought, also shown in other studies (Peltoniemi et al., 2016) might have persisted

after the brackish water inflow because re-establishment was likely hindered by competition for substrate with sulfate reducing bacteria (Schönheit et al, 1982; Scholten et al., 2002; van Dijk et al., 2019). In deeper layers, however, the increase in absolute methanogenic abundance (Fig. 5b and f) might result from a lack of competition between SRB and methanogens for substrate. This could have two reasons: 1) There was enough labile litter available after the drought due to plant die-back (Hahn-Schöfl et al., 2011) and different microbial processes are taking place simultaneously or 2) methanogens did not use organic

compounds such as acetate, but rather methylated compounds (Söllinger and Urich, 2019) or hydrogen and $CO_2$. In this context, the microbial community data and carbon isotopic signatures of $CH_4$ and DIC suggest a relative increase in methanogenic $CO_2$-reduction which potentially benefits from an increase in non-methanogenic $CO_2$ production.

Similar to the isotopic values and $CH_4$ concentrations, the molecular microbial data provide no evidence for substantial methane oxidation in the peat after the brackish water inflow. Although we detected some taxa associated with anaerobic

methane oxidation, specifically *Candidatus* Methanoperedens (ANME 2d) and ANME-3, their abundance was very low, especially on the level of transcripts. This holds in particular for groups known to be involved in sulfate-driven anaerobic methane oxidation (AOM). However, anaerobic methanotrophic archaea are known to be slow growing (Nauhaus et al. 2007; Holler et al., 2011; Knittel et al., 2018) and seem to require stable environmental conditions (Ruff et al., 2016). Peatlands,



rewetted ones especially, are generally highly dynamic systems with regard to hydrology and redox conditions and the supply
of electron acceptors, mostly sulfate, after the inflow may not have been suffice for AOM communities to establish. Aerobic
methanotrophs, on the other hand, may be hampered in their activity by the standing water above surface and lack of oxygen.
Consequently, their population could have become inactive without any major changes in population size and community
structure.

Finally, the brackish water inflow could have been associated with an introduction of marine-derived aerobic and anaerobic
methanotrophic taxa. A measurable change in community composition through this, however, was not observed which is
further supporting that methane oxidation was no relevant process after the storm surge in the peat unlike it was in the period
after the drought in 2018 (Unger et al., 2021). As discussed earlier, though, methane oxidation most likely occurred in the
standing water above the peat given the substantial drop in methane emissions despite the fact that methanogenesis seemingly
occurred besides alternative anaerobic pathways of carbon respiration, mostly sulfate reduction. Methane oxidation in the
water column was, however, beyond the scope of our study. Therefore, in the future, it seems advisable to include the above
peat layers, namely, the open water and the fresh litter in similar studies.

**5 Conclusion**

Brackish water inflow led to an increase in electrical conductivity and sulfate concentrations in the surface and pore water of
a coastal fen that had originally been rewetted with freshwater. This resulted in a recharge of sulfate concentrations in the
upper pore water layers and a homogenization of the microbial community composition and abundance along the depth
profiles. Trace gas concentrations show an overall decrease in methane and $CO_2$ concentrations after the brackish water inflow.
Isotopic signatures unexpectedly suggest increased formation of more $^{13}C$ depleted $CH_4$ and DIC, indicating ongoing
methanogenesis though shifted towards more methanogenic $CO_2$-reduction and non-methanogenic $CO_2$ production. At the
same time no evidence for substantial aerobic and anaerobic methane oxidation was detected in the peat. Furthermore, sulfate
reducing bacteria (SRB) increased in overall abundance and diversity throughout the whole peat profile. Presumably, the
presence of sulfate helped SRB to establish a large community in the peat soil, although many members of this large group
had been already present in locally confined high-sulfate environments during the drought. It remains unresolved, however,
why methane emissions decreased to a new minimum since rewetting more than a decade ago, while methanogenic absolute
abundances and methane concentrations overall did not change or even decreased. Possibly methane oxidation took place
within the water column or the fresh litter on the ground surface above the peat which was, however, outside of the scope of
this study.

In conclusion, the inflow of brackish water into a freshwater rewetted, highly degraded coastal fen likely contributed to further
reduce methane emissions following a drought in the preceding year. The sequence of drought and storm surge profoundly
altered $CH_4$ emissions and underlain microbial communities although at the same time the precedent drought seemingly
interfered with the effect of the inflow. Rising sea levels (and stronger storm surges) due to climate change are likely to cause

an increase in the frequency of brackish water inflow events into coastal peatlands. This will affect the sulfate-methane dynamics in these systems and thereby change their biogeochemical cycling processes and most likely decrease methane emissions.

## Data availability

All raw data can be provided by the corresponding authors upon request.

## Authors contribution

GJ, SL and FK developed the idea and concept of the research project; CNG, GJ, SL, VU and FK planned data collection, manuscript structure and the research focus; CNG, A-KJ, EDR performed the field campaigns and subsequent data analysis and corrections; CNG, A-KJ, DO, IS and LW performed lab analysis; SY did bioinformatic analysis, CNG and GJ did statistical
analysis and created the figures; CNG, K-HK, SL, GJ, VU, FK and MJ did data interpretation; CNG summarized all data and wrote the manuscript with the help of GJ, A-KJ, EDR, MJ, SY and K-HK, who wrote parts of the method section. GJ, SL, FK, VU, K-HK, MEB, JK, MJ and EDR reviewed and edited the manuscript.

## Competing interests

The authors declare that they have no conflict of interest.

**Acknowledgements**

We thank Anke Saborowski, who helped to a large extent in the laboratory during molecular extraction, quantification and preparation for sequencing of microbial data. Further, we thank Jan Axel Kitte and Oliver Burckhardt for their kind assistance in the laboratory during GC measurements. Special thanks also go to Daniel Brüggemann, who performed isotopic analysis at the University of Münster and Evelyn Bolzmann, who carried out the loss on ignition measurements at the Soil Physics
department of the University of Rostock. In addition, we thank Joachim Hofmann and Birgit Schröder for their technical support and kind cooperation, Dr. Anke Günther for her help with the $CH_4$ flux calculations and the provision of the according R script, Dr. Vytas Huth for his advices regarding field equipment, sampling design and approaches and Dr. Sate Ahmad for interdisciplinary discussions and his QGIS expertise.





**Financial support**

This work was supported by the German Research Foundation (DFG) within the PhD graduate school "Baltic TRANSCOAST" (GRK 2000/1) in the framework of the Open Access Publishing Program.

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

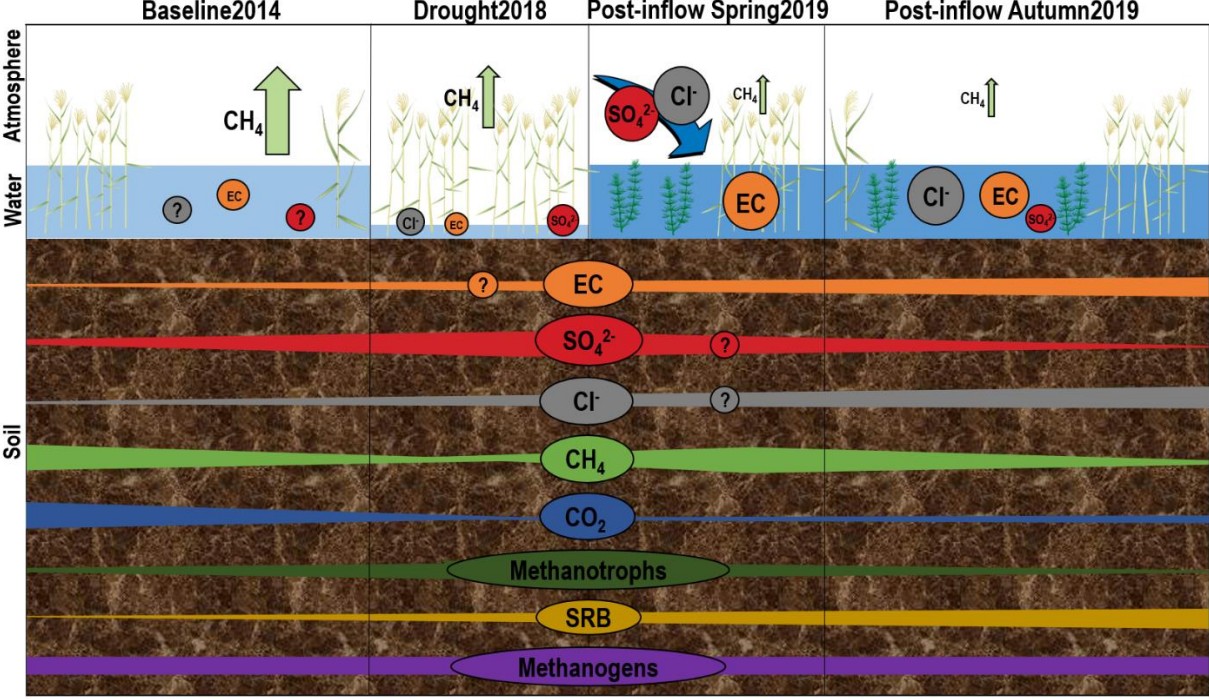

**Figure 1: Schematic development of geochemistry, greenhouse gases and microorganisms under different environmental conditions**
**throughout time at three different compartments discussed in this study: atmosphere, surface water and peat soil. Patterns were derived from annual budgets of CH₄ fluxes (green arrows, Koebsch et al., 2020) and from concentrations of surface and pore water components, averaged over all locations and depth sections. Note that CH₄ and CO₂ patterns show tendency derived from peat soil concentrations, not from isotopic signatures of δ ¹³C. Schematic microbial changes are based on absolute counts of qPCR results. The design of plants and other symbolic depictions was inspired and partly extracted from the media library of the Center for**
**Environmental Science, University of Maryland (https://ian.umces.edu/media-library/symbols/#download).**



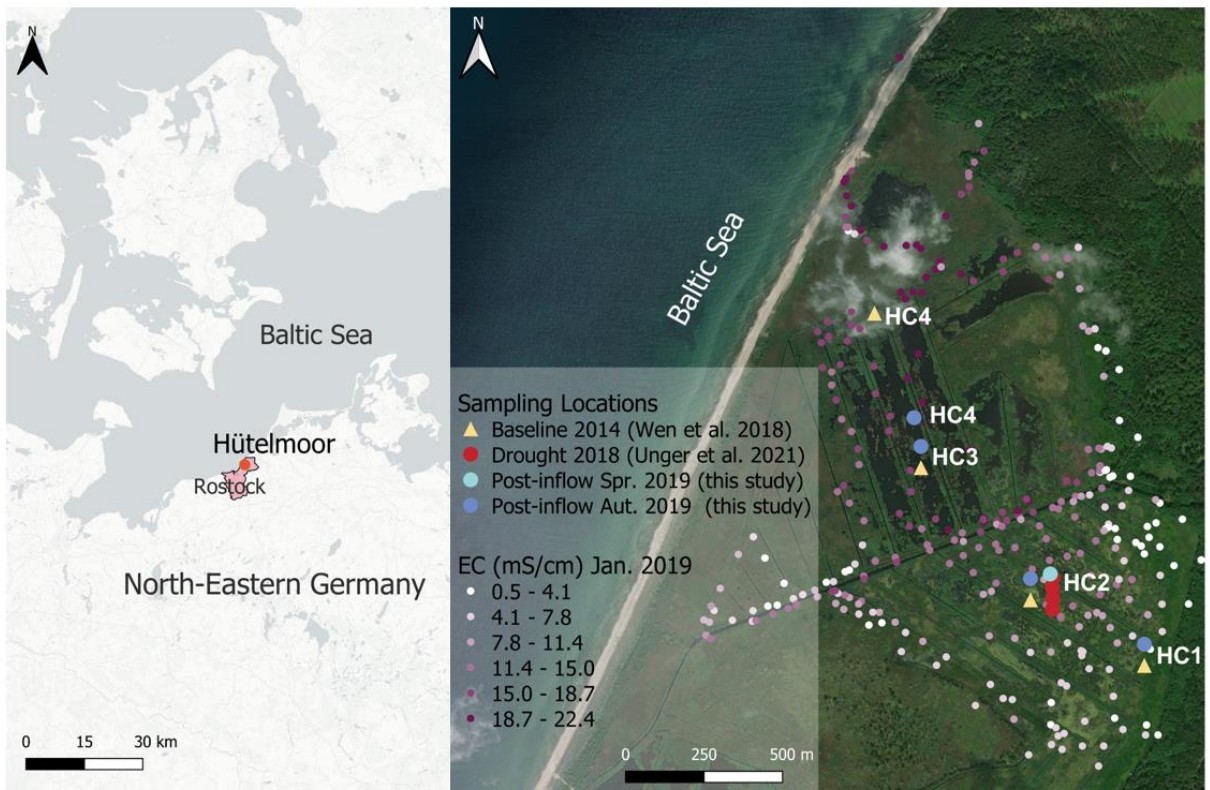

**Figure 2: Location of the study site in North-Eastern Germany (left) and sampling locations HC1-4 within the study site "Hütelmoor" (right). Exact locations of baseline sampling in 2014 are shown in yellow, drought sampling in 2018 in red, post-inflow in spring 2019 in light-blue and post-inflow sampling in autumn 2019 in dark-blue. Due to technical reasons, location HC4 had to be shifted post-inflow towards south from its original (Baseline2014) position. Electrical conductivity (EC) values from January 2019, shortly after the inflow of brackish water, are shown in different shades of purple and ranged from 0.5 to 22.4 mS/cm. Location map was drawn in GQIS, version 3.22.4 and base map were extracted from: https://a.basemaps.cartocdn.com/light_nolabels/{z}/{x}/{y}@2x.png, https://www.geoportal-mv.de/gaia/gaia.php and http://server.arcgisonline.com/arcgis/rest/services/World_Imagery/MapServer.**








**Figure 3: Compilation of depth profiles for pore water variables; Letters a) to d) and i) to l) show comparison between Baseline2014 (n=42) and Post-inflow Autumn2019 (n = 26-32) at locations HC1-4; e) to h) and m) to p) show comparison between Baseline2014 (n = 12), Drought2018 (n = 24), Post-inflow Spring2019 (n = 8) and Post-inflow Autumn2019 (n = 8). Colors represent the different sampling campaigns and symbols show different sampling locations. The lines depict a span=0.5 LOESS smooth along the data points and are meant to guide the eye. The shaded areas represent the respective confidence interval of 95 % according to standard**
**errors of the models. Colorblind-friendly color palette "4-class RdYlBu" was used from: https://colorbrewer2.org/?type=diverging&scheme=BrBG&n=4#type=diverging&scheme=RdYlBu&n=4**

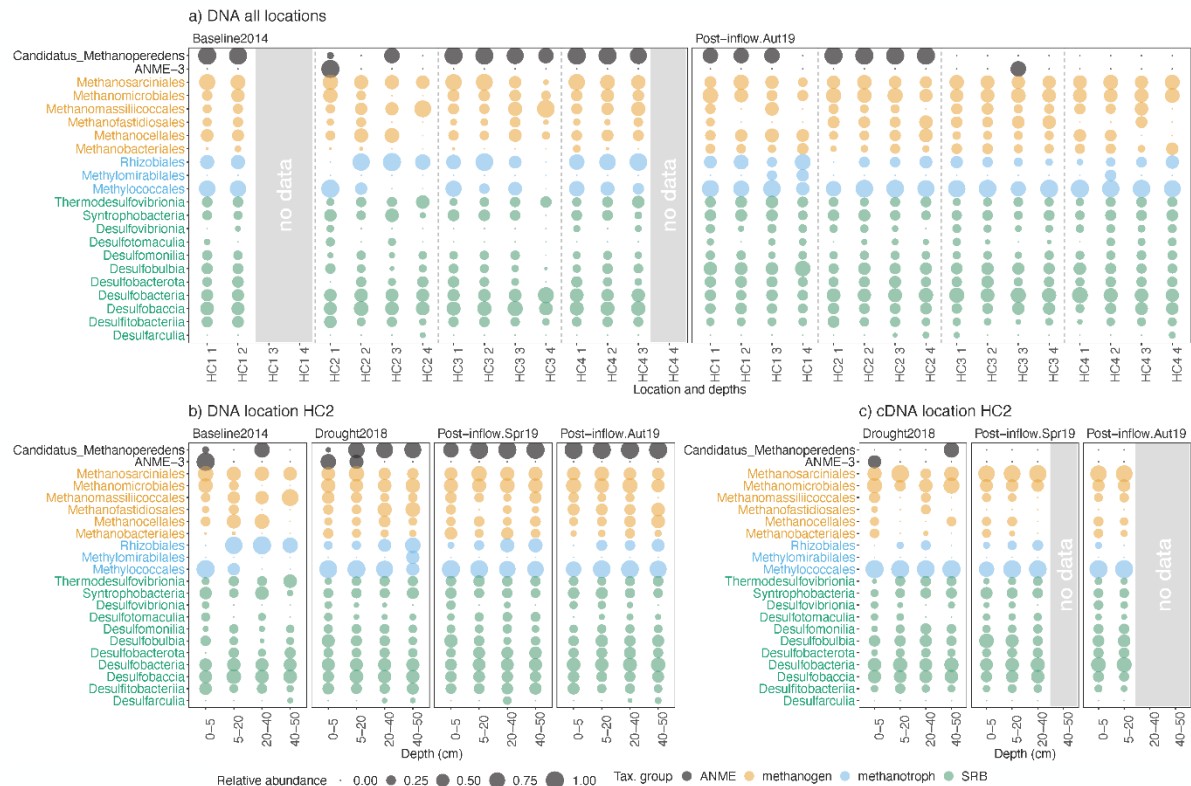

**Figure 4: Bubble plots showing the microbial community composition and relative abundances from all sampling locations along the surface water salinity gradient (a) and the sampling location HC2 (b and c). On the y-axes the taxonomical groups on order**
**(methanogens, methanotrophs), class (sulfate reducing bacteria (SRB)) and genus level (anaerobic methanotrophic archaea (ANME)) are displayed. The x-axes show a) the locations HC1-4 and sampling depths, where codes correspond to the following depths: 1 = 0-5, 2 = 5-20, 3 = 20-40, 4 = 40-50 cm and b) and c) the depth in cm. Coloring reflects the different microorganism groups. Circle sizes represent relative abundances (sqrt transformed) of different taxonomic groups derived from a, b) DNA- and c) cDNA-based sequencing. Note, that groups are not adding up globally, but sum up to 100% within each group (methanogens,**
**methanotrophs, SRB, ANME).**



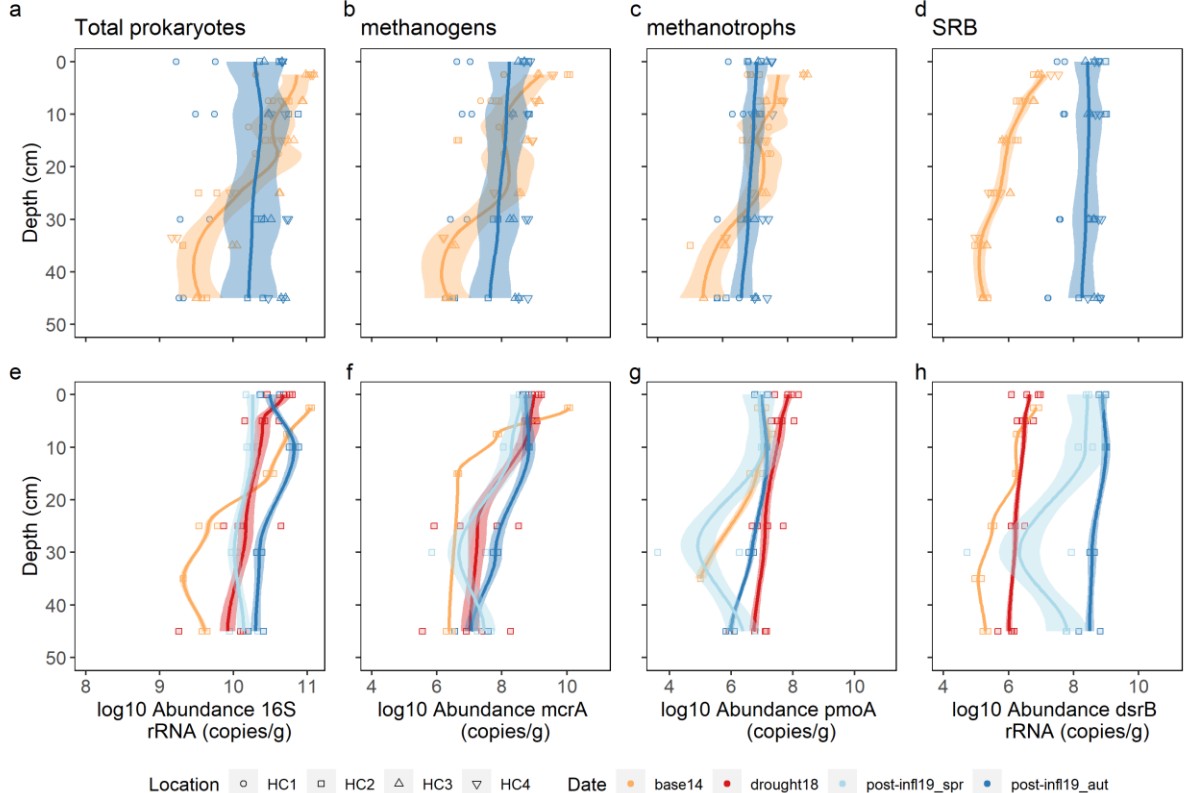

**Figure 5: Depth profiles a) of locations HC1-4 and b) location HC2 showing log10 abundances (copies/ g dry soil) of functional genes (16S rRNA, *mcrA, pmoA* and *dsrB*), derived from qPCR analysis. Sample sizes differ between top and bottom plot and are as follows: HC1-4 in Baseline2014: n = 34 - 42, Post-inflow Autumn2019: n = 32; HC2: Baseline2014: n = 7-12, Drought18: n = 16, Post-inflow Spring2019: n = 8, Post-inflow Autumn2019: n= 8. Different colors visualize different sampling dates. Trend lines were estimated using LOESS with a span of 0.5 and are meant to guide the eye. Shaded areas show confidence interval according to standard errors. Confidence interval for all locations (a-d) is 95%, confidence interval of location HC2 (e-h) was set to 50%.**





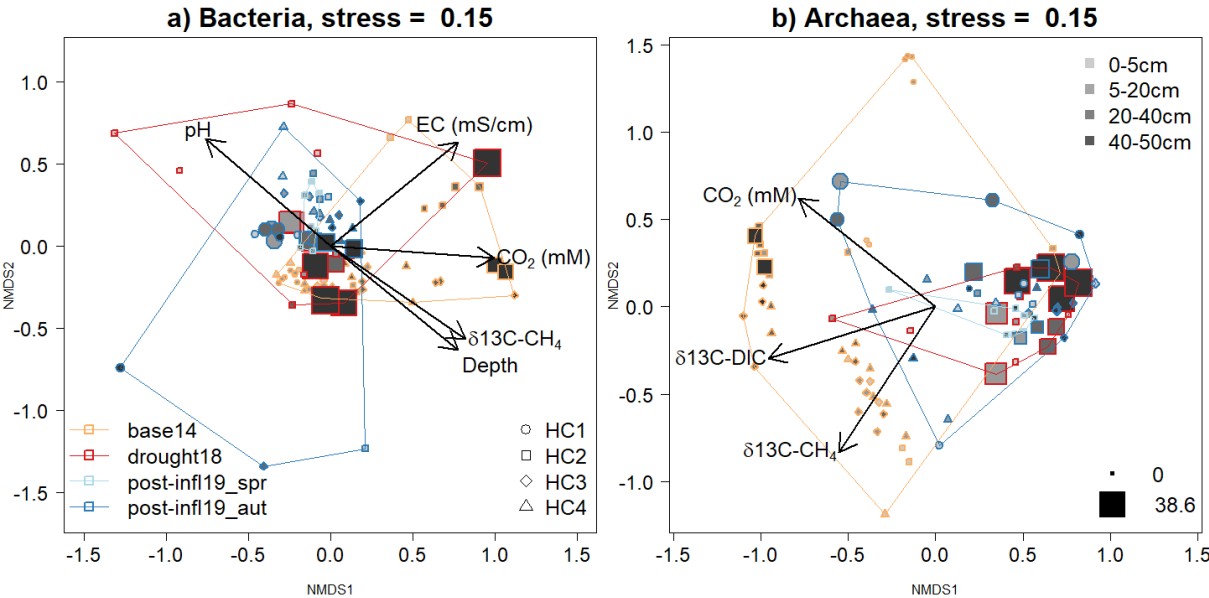

**Figure 6:** NMDS ordination on bacterial and archaeal community composition according to sampling campaign (polygons, for color-coding see legend bottom left in a), sampling locations (for symbols of HC1-4 see legend bottom right in a), depths (for grey shades see legend top right in b) and sulfate concentrations (minimum and maximum values in mM represented by symbol sizes, see legend bottom right in b). Proximity of colored symbols can be interpreted as similarities in bacterial and archaeal community composition. Arrows indicate the direction of change in environmental variables (only those variables are shown that showed significant correlation to the domination configuration).