# Peer review of "EFFECTS OF BRACKISH WATER INFLOW ON METHANE CYCLING MICROBIAL COMMUNITIES IN A FRESHWATER REWETTED COASTAL FEN"

_EGUsphere, 2022_

## Author Comment (AC2)

| Accession or Name | | | | Release status | | Maximum rows | |
|---|---|---|---|---|---|---|---|
| | | | | | ▼ | 100 | ▼ |

Download all results (https://www.ebi.ac.uk/ena/submit/report/projects/PRJEB52161?max-results=10000000&
format=csv&
token=eyJhbGciOiJSUzI1NiJ9.eyJwcmluY2lwbGUiOiJXZWJpbi00MDQzOCIsInJvbGUiOltdLCJleHAiOjE2NTQwNDMzMzk
HxVv85Ey8gb4a8yNBaqRu9tRnXHG9ZXL9DLilPyUrUqCJyBhmigYl3pGFJ2ZwcWB9hM_fQUp1cp-zfQ-
ZJHUL31_1nDQCgqsXmcZSE9WBUWT1NBIqIBcAaIPBVyqbeLuokKT2MB3MhJ4H601zjtuPiQ50PpoOSq3rlmnJxalKl_mii
fiMCDRDb25XB-ykl3s0Ldfl0MUQ0c4AghjrWyp2mze9cZ00y8vBIU3vMv_hjDP-mSVmNg)

| Accession | Secondary Accession | Title | Submission date | Release date | Status | Action |
|---|---|---|---|---|---|---|
| PRJEB52161 | ERP136848 | Microorganisms as drivers of CH4 production and exchange in coastal fens after brackish water inflow | 6th Apr 2022 | 31st Oct 2022 ✏ | Private | |

---

## Author Response (AR1)

Reviewer 1

Dear Reviewer 1,

Many thanks for your very constructive feedback on our manuscript. We think, most of your comments mentioned will find their way into the manuscript and thus, you helped us very much in improving it. Yes, unfortunately we were not able to take a more detailed look into the microbial community other than we did. Thank you for acknowledging the broad range of data covered. To answer you comments in detail, we will re-post your comment and our reply right below.

Major comments:

1. In the discussion, the possible reasons behind the geochemical patterns are considered in great detail and very well, but I was missing more broader context. What do these results mean outside this specific system and outside these specific sampling points? What new was learned that could be generalized to the effect of drought and brackish water inflow in other wetlands with brackish influence?

Reply: Thank you very much for this important comment. While discussing the issue among authors we realized that we were indeed somewhat missing the broader picture especially in the discussion. Therefore, we suggest to add the following paragraph after the sentence in l. 557:

"So, what does this mean for a broader context? The whole peatland was affected by a single storm surge and the resulting brackish water inflow. Such events are likely to happen more frequently and possibly more intensely in the future in the investigated site and in many low lying peatlands as a consequence of global warming induced sea level rise (Jurasinski et al., 2018). In parallel, as temperatures increase and weather patterns are getting more extreme, drought periods in peatlands may occur more often in the future. Thus, we were able to study possible future events, rendering the results exemplary for other coastal peatlands. The change from drought conditions to brackish water inflow might even trigger similar process chains in non-rewetted, still drained fens, since their normal is the dry situation.

Brackish water inflow is sometimes, also by some of the authors, discussed as a possible way to reduce methane emissions after rewetting of peatlands, even if they are not intentionally rewetted as a natural-based solution for climate change mitigation. However, although the sulfate input and/or activation we have seen, seems to have been beneficial because it leads to lower methane emissions, salinization is also seen as a dangerous threat to many coastal ecosystems. In addition, sulfate might lead to higher peat mineralization rates (Zak et al., 2019) and the produced $CO_2$ could outweigh the positive effects of lower methane emissions in the long-term. Therefore, further research in a variety of shallow coast peatland ecosystems is necessary to draw final conclusions. Since these complex ecosystem effects are hard to investigate in experimental studies, this calls for a network of long-term monitoring sites."

In addition, as a reaction to specific comment no. 11, we will also change the sentence in lines 609-610, so that the changed role of the water column, is better acknowledged. Before conducting the study, we expected the water column to be a source for methane-production, but we suspect it to rather fulfill the function of methane oxidation. We think, that this has also wider meaning outside of the specific case study.

In the conclusion however, we consider the broader context sufficiently covered and would prefer not to change the text, also in order to avoid the concluding paragraph to get too long.

2. Please mention the accession number for the nucleotide sequences in the main manuscript (now it is mentioned in Table S2). I can't find anything with the accession number PRJEB52161- are the data not public yet?

Reply: Yes, you are totally correct, the accession numbers will be included into the text. The reason for the incomplete statement in the manuscript under "Data availability" is that the manuscript submission was done before the uploading of the data. We are planning to include the following paragraph into the manuscript in l. 680 and replace the sentence, which is currently there:

"The data for all 97 post-inflow samples have been deposited in the European Nucleotide Archive (ENA) at EMBL-EBI under accession number PRJEB52161 (with sample accession numbers ERS11559347-ERS11559443). Baseline2014 data can be found at EBI under the BioProject PRJNA356778 (accession numbers are SRR5118134-SRR5118155 and SRR5119428 - SRR5119449) and Drought2018 data were deposited at ENA under BioProject accession number PRJEB38162 (sample accession number ERS4542720-ERS4542735, ERS4542752-ERS4542767, ERS4542784-ERS4542800 and ERS4542822-ERS4542837). Depth profile data can be provided by the corresponding authors upon request and will be uploaded to the Pangaea data base in the near future.".

And yes, the data are not public yet, but were submitted on 6th April 2022. The predicted release date is 31th October 2022, please also see the attached record below. However, the data might be public even before that date.

Specific comments:

1. line 156 What is meant with 'for better comparison' here? Same sampling time or something else?

Reply: This is mostly about a spatial effect. The drought sampling only took place at location HC2 and so did the Spring post-inflow sampling 2019. We wanted to say that at location HC2 there is a higher temporal resolution, with two additional samplings (drought and spring 2019). Unlike the other stations, where sampling only took place in 2014 and Autumn 2019. But in addition, yes, this is also about the sampling time. The drought sampling took place in August 2018, so while including May and November 2019 sampling, we relativize the seasonal influence slightly.

To make it easier to follow our thoughts here, and to better reflect our intentions, we suggest to change the sentence to: "Soil cores and pore water samples were also taken on May 16[th], 2019 ("Post-inflow Spring2019") at one of our sampling locations (HC2, see Fig. 2) for better comparison with the previous drought study (Unger et al., 2021) in order to increase the temporal resolution at this common location.". The following sentence ("This sampling was, however, only done at one of our sampling locations (HC2, see Fig. 2) ") will be deleted.

2. l. 274-275 Should the primer concentration be 0.5 uM instead of 0.5 mM? 0.5 mM would be an unusually high concentration.

Reply: Yes, that is totally correct, it should be μM. Thank you very much for the careful read. We will change the unit accordingly.

3. l. 275 What was the final volume of the PCR reaction?

Reply: The final volume was 25 µl. We will change the sentence to reflect this detail to: "For the PCR (Thermal Cycler, T100, Biorad, Feldkirchen, Germany) we added PCR-Buffer, 1.25 U OpitTaq DNA Polymerase, 0.2 mM dNTP, 0.5 mM $MgCl_2$ and 0.5 µM of each primer to 5 µl purified sample and filled the mixture to a final volume of 25 µl using sterile water."

4. l. 292 Please mention the concentration of the primers.

Reply: Ok, we will change the sentence to: "According to the in-house protocol, we used 10 µl of SYBR Green, 0.08 µl of each primer (with a concentration of 100 µM each), 5.84 µl sterile water and 4 µl template per reaction."

5. l. 480-482 Please be careful when directly comparing the results of two different qPCR assays. We can't know if the abundances are affected by primer biases etc.

Reply: Well, actually the assays were especially prepared in a way that they are indeed comparable to one another, since we used standards with known gene copy numbers for each gene, so absolute abundances should be correct. Please see also Wen et al. (2018) and Unger et al. (2021) for method comparison. However, your comment makes us aware that a single comparison between *mcrA* and *pmoA* is not of great relevance for our manuscript, because we mainly focus on the comparison of the same gen within different time frames. So, we would also agree to delete the sentence or change the wording as the following: "After the brackish water inflow, absolute *mcrA* gene abundances of DNA-based analysis were substantially higher compared to *pmoA* abundances, which is also reflected in the cDNA-based abundances from location HC2 (Table S1)."

6. l. 496-498 I understand what is meant here, but please try to rephrase this sentence taking into account that the environmental variables are properties of the soil samples, not bacteria (for example that in the bacterial ordination, these samples were associated with higher EC etc.).

Reply: Thank you for this comment. We will change the sentence accordingly. It could read as following: "In the bacterial ordination, the Baseline2014 samples were associated with slightly higher EC and $CO_2$ concentrations and with more enriched $^{13}C$ in $CH_4$ (see post-hoc fit arrow in Fig. 6a) compared to the other sampling campaigns."

7. l. 522 Please consider reminding the reader here if the sampling point HC2 is closer to Baltic Sea or further inland.

Reply: This is a good point indeed. Thank you. To acknowledge this comment, we will change the sentence in the previous lines 518-519 to: "Instead, two zones of different brackish impact, separated by the main ditch, formed with higher EC concentrations close to the Baltic Sea (HC3 and HC4) and lower EC concentrations further inland (HC1 and HC2, see also Fig. 2)."

8. l. 539-541 I'm not sure why but I'm having difficulties following this sentence. Please consider if you can clarify the main point of sentence or its connections to what is said above.

Reply: To make clearer what we wanted to state here, we will try to separate the information into two sentences: "Despite the fact that the locations differed in pore water biogeochemistry, the shift from freshwater to brackish conditions was clearly visible. This

is especially true, because sulfate, chloride and EC levels showed an approximation of the freshwater-influenced upper part and the partly brackish-influenced deeper pore water (HC2) after the inflow (Fig. 3b, c and d)."

9. l. 554 What is meant with 'drought-induced salinization'?

Reply: We refer to drought-induced salinization in the introduction in l. 99. It is a rather broad term, used e.g. in Chamberlain et al. (2020) to describe the increase of salinity (measured in PSU) during drought conditions. Presumably they used the term synonymously to sulfate to describe sulfate-enrichment during drought, resulting from the re-oxidation of sulfide under aerobic conditions. In the discussion, we wanted to draw a connection with the introduction, but this time emphasizing, that chloride cannot, like sulfate increase simply because of dry conditions, but was most likely transported from the sea. So, because sulfate is a difficult proof of the brackish-water inflow, we used chloride additionally to support our hypothesis that brackish water inflow indeed happened and that sulfate did not only increase because of the aerobic conditions during the drought.

10. l. 579 Please clarify here which time point has the lower values.

Reply: Yes, of course. We will change the sentence accordingly to: "The decrease of $\delta13C$-DIC between the baseline sampling in 2014 and autumn 2019 sampling post-inflow indicates an increase in non-methanogenic $CO_2$ production (Fig. 3l)."

11. l. 609-610 Please check if you can clarify this sentence. I'm especially having trouble with the word 'changes' on l. 610.

Reply: We tried to clarify what we mean by rephrasing the sentence. We suggest the following: "Therefore, carbon cycling might have changed after the complex impact of drought and subsequent brackish water inflow from well-known patterns, turning the usual role of the water column from a source of methane into a methane emission avoidance function in the investigated ecosystem."

12. l. 1223-1224 I see from the R markdown file (thank you for including this file!) that the arrows for the environmental factors come from envfit, but this should be mentioned in the methods section too.

Reply: Yes, good point. We suggest to add the following sentence between the sentence in l. 333 and l. 334: "We used the function envfit of package vegan (Oksanen et al., 2020) in order to add environmental variables on the NMDS ordination configurations."

Minor technical or language comments:

l. 48 Open the abbreviation 'GHG'.

Reply: Thanks, will be changed as suggested.

l. 151 Open the abbreviation 'EC'.

Reply: Thanks, will be changed as suggested.

l. 164 Open the abbreviation 'IC'.

Reply: Thanks, will be changed as suggested.

l. 169 0,45 -> 0.45

Reply: Thanks, will be changed as suggested.

l. 273 'Specific' could be a better word here than 'precise'?

Reply: Yes, you are right, "precise" alone may be a bit off here. However, we would like to change the wording to "more precise". We wanted to emphasize the fact that universal primers were less suitable to detect archaea relative abundances precisely enough and therefore archaeal primers were used.

l. 424 and elsewhere: Methanosarciniales -> Methanosarcinales

Reply: Thanks, will be changed as suggested.

Reviewer 2:

Dear reviewer2,

Many thanks for your feedback, which we find constructive and very valuable. Especially, we thank you for spotting that the methods used to compare microbial community abundance and composition with earlier studies were not well described. First of all, we want to assure you that the methods in the different studies are comparable and in fact we put a lot of emphasis on comparability of the approaches being aware of potential flaws associated with DNA extraction methods, PCR protocols, sequence data processing etc. All microbial analyses were done in the same lab, using the same DNA and RNA extraction kits and the same primer combinations for the qPCR. The PCR used to amplify and tag the individual samples for sequencing was also done in the same way as the previous sister study. Only the specific bacterial primer used earlier (S-D-Bact-0341-b-S-17/S-DBact-0785-a-A-21) was replaced by a universal primer targeting both bacteria and archaea with, however, the same resolution for bacteria. In all studies, archaea were amplified separately to get an in-depth analysis of archaeal community composition. When re-editing the manuscript, we will try to make the comparability of the three studies clearer by including the following statements into the manuscript:

l. 144: "The microbial analysis was conducted in the same laboratory and strictly followed the same protocols regarding DNA and RNA extraction and the usage of the primer combinations during sequencing and qPCR. Minor adaptations due to improved technologies are marked accordingly in the relevant subchapter of the method section."

l. 273: "Please note that Wen et al. (2018) and Unger et al. (2021) used a specific bacterial primer combination (S-D-Bact-0341-b-S-17/S-DBact-0785-a-A-21) instead of the universal primer we used here. We decided for the universal primer, because it has equal resolution for bacteria, but covers both, bacteria and archaea providing some back-up of the sequencing and qPCR data."

l. 307: "All sequencing reads, including those from Wen et al. (2018) and Unger et al. (2021) were merged into a common ASV file which provided the basis for all following analyses."

We would also like to draw your attention to existing lines in the manuscript, where we describe the normalization process of all data used (Wisconsin double standardization) to create the bubble plots (Fig. 4) in l. 328 and to create the NMDS ordination (Fig. 6) in l. 333.

Regarding your question concerning the data depository, please see our reply to reviewer1's second major comment:

'Yes, you are totally correct, the accession numbers will be included into the text. The reason for the incomplete statement in the manuscript under "Data availability" is that the manuscript submission was done before the uploading of the data. We are planning to include the following paragraph into the manuscript in l. 680 and replace the sentence, which is currently there:

"The data for all 97 post-inflow samples have been deposited in the European Nucleotide Archive (ENA) at EMBL-EBI under accession number PRJEB52161 (with sample accession numbers ERS11559347-ERS11559443). Baseline2014 data can be found at EBI under the BioProject PRJNA356778 (accession numbers are SRR5118134-SRR5118155 and SRR5119428-SRR5119449) and Drought2018 data were deposited at ENA under BioProject accession number PRJEB38162 (sample accession number ERS4542720-ERS4542735, ERS4542752-ERS4542767, ERS4542784-ERS4542800 and ERS4542822-ERS4542837). Depth profile data can be provided by the corresponding authors upon request and will be uploaded to the Pangaea database in the near future." '

Line comments:

Line 33: I find this sentence a bit confusing to read. Perhaps remove "also"?

Reply: Thank you for your comment. The "also" was there to emphasize the fact that the drought lead to the results we see IN ADDITION to the brackish water inflow. We would like to suggest the following change: "We found that both, the inflow effect of brackish water and the preceding drought increased the sulfate availability in the surface and pore water."

Line 46: do the authors intend "loose" instead of "lose"?

Reply: Yes, thank you very much for the careful read. We meant "loose" and will change the text accordingly.

Line 112-114: sentence a bit confusing to read, do the authors intend "Thus" instead of "This"? In particular the " and, therefore can explain" is causing some befuddlement.

Reply: Thank you very much for the hint. We meant "this", because it links to the previous sentence and refers to the increase of SRB at the expense of methanogens. This increase together with an increase of ANMEs should lead to a decrease of methane production. We will try to improve the sentence and suggest the following: "The increase of SRB in conjunction with an anticipated increasing abundance of sulfate-dependent anaerobic methanotrophic archaea (ANMEs) should decrease methane production and, therefore can explain the reported decrease in methane emissions."

Line 198: Can you please clarify how samples were kept cold and if they were kept anoxic? I would be concerned that collecting the soils, cooling them on ice (in a cooler?), then later placing the samples in a -80 freezer would not preserve the RNA as it was in the field, especially as there may have been an influence of O in the previously anoxic depths as the cores were stored in falcon tubes. Please indicate the length of time between sampling and

freezing, if greater than a few hours, then the RNA data may be more reflective of the storage conditions and not the in-situ conditions.

Reply: This, indeed, a well justified concern. Samples for RNA extraction (only one site, HC2, out of four) of this study were stored in a cooler box first and frozen within three hours after sampling. We agree that we cannot exclude some level of RNA degradation during this period unlike in the previous study by Unger et al. (2021), where samples were stored in a dry shipper immediately after sampling. Nevertheless, RNA extraction and cDNA synthesis were timely and successful. Given that samples were stored at very low temperatures as soil samples, i.e. in their original matrix, a shift in community composition also of the active fraction in such a short time is unlikely. This corresponds with our results which show a large level of similarities between the two cDNA datasets. However, taking your concern into account, we decided to delete parts of the manuscript stressing taxa that were less represented in our study since in fact such a decrease may have resulted from decay of RNA rather than from real changes in the active community. We will therefore delete the sentences in the following lines:

l. 462: "Methylomirabilales were not detected in the cDNA-based extractions".

l. 465-466: "However, classes such as Syntrophobacteria and Desulfobulbia showed higher cDNA-based abundances only after the drought in Post-inflow Sping2019 in the surface peat layers."

l. 468-470: "These findings can however not be confirmed with the data on cDNA-based abundances, suggesting no active role of *Candidatus* Methanoperedens except during Drought2018 in the deepest peat layer at HC2."

l. 472-473: "According to the cDNA analysis, active ANME-3 were little abundant in the surface peat layers during the Drought2018 (black, Fig. 4c)."

Since there is very little data on the active communities in rewetted fens and since the cDNA data do not form the core of our study but rather serve as additional information, we would like to leave the cDNA in the manuscript. In order to avoid overinterpretation of the data, we decided to make the reader more aware of the different preservation procedures, though. Therefore, we will state the following in the caption of Fig. 4 in l. 1210: "Please also note that preservation methods differed slightly between the studies."

Line 274: which PCR buffer? What was the final reaction volume? Was the same amount of DNA added to each reaction?

Reply: Many thanks for these considerate additions. We used 10x Pol Buffer C by OptiTaq DNA Polymerase (Roboklon). 50 µl was the final volume and 5 µl were used from each sample. Also considering comment no. 3 by the first reviewer, we suggest to change the sentence to: "For the PCR (Thermal Cycler, T100, Biorad, Feldkirchen, Germany) we added 10x Pol PCR-Buffer C (OptiTaq DNA Polymerase, Roboklon, Berlin, Germany), 1.25 U OptiTaq DNA Polymerase, 0.2 mM dNTP, 0.5 mM $MgCl_2$ and 0.5 µM of each primer to 5 µl of the purified sample. Using sterile water, we filled the mixture to a final volume of 50 µl." We also apologize for the error (final volume of 25 µl) made in the reply to the first reviewer. We checked again and found that 50 µl was the correct final volume.

In case your comment addressed the comparability of the amount of DNA between the three studies, we want to emphasize that the patterns between copies per g soil or copies per ng DNA (which we calculated additionally) are consistent and show the same trends, so that the unit (copies/ g soil) represents the amount of DNA contained.

Line 292: Was the same amount of DNA added to each reaction? How was this normalized? What is the final reaction volume? What was the primer concentration?

Reply: Thank you for pointing this out. The same volume of purified DNA (4 µl) was used for each reaction. Since absolute copy numbers per gram soil were calculated, it was not necessary to use the same weight of DNA as starting material. Especially, as shown above, the normalizations against gram of soil and ng of DNA reveal the same trends. In reaction to your comment about the final reaction volume, we would modify the sentence further in addition to the changes suggested by reviewer1 (primer concentrations): "According to the in-house protocol, we used 10 µl of SYBR Green, 0.08 µl of each primer (with a concentration of 100 µM each), 4 µl template per reaction and 5.84 µl sterile water, resulting in a total final volume of 20 µl."

Line 296: standard curve was based on a series of dilutions of what? Please indicate brief methods, even though detailed in the sister studies.

Reply: Yes, we will give some brief details on the dilution method and suggest the following detail addition: "The standard curve was typically based on a series of dilutions of known numbers of concentrations in the range of $10^3 - 10^8$ copies as specified in Winkel et al. (2018), with starting concentrations being $2.5 \times 10^8$ for 16S rRNA, $2.9 \times 10^7$ for *mcrA*, $3.2 \times 10^7$ for *pmoA* and $6.69 \times 10^7$ for *dsrB*."

Line 296: How were gene copy numbers normalized to the amount of soil used for DNA extractions? Based on the x-axes in Figure 5 this appears to be the case, but please clarify in the text.

Reply: Thank you very much for this important feedback. All absolute gene copy numbers (copies/µl) were multiplied by the final DNA extraction elution volume (50, 60 or 100 µl), the dilution factor (mostly 10 or 100, sometimes 1) and divided by the initial fresh weight of the individual soil sample. In order to normalize the different soil water content values, a dry weight factor was determined (wet weight/dry weight) and multiplied with the gene copy numbers to get the gene copies number per g dry soil. We intend to add the following after l. 299: "All absolute gene copy numbers are given per gram dry soil and were calculated by normalizing them over their initial fresh weight taking into account a dry weight factor, the elution volume and the dilution factor. For better visualization, we log10 transformed the data."

Line 330: how were ordination vectors constructed?

Reply: Do you mean, the environmental variables? We used the function envfit from vegan package. Thanks to reviewer1, we intend to add the following sentence between the sentence in l. 333 and l. 334: "We used the function *envfit()* of package vegan (Oksanen et al., 2020) in order to add environmental variables on the NMDS ordination configurations."

Line 386 "Drought"

Reply: Thank you very much! We will correct the tipping error.

Line 371 (and elsewhere): please indicate if these are average +/- standard error, perhaps in the methods? Or indicate the first time mentioned.

Reply: Yes, good point. We suggest to modify the sentence in l. 341 as follows: "To display average values for different subgroups (usually mean values with standard error if not indicated otherwise), we used the psych package (Revelle, 2020)."

Line 425 (editorial comment, can be ignored): to my eye, the colours appear more yellow than orange in the online version.

Reply: Thanks for the comment. Colours may appear different to every individual reader. Since we assume that the chosen colour palette will make sure that colour can be differentiated from each other (even for people with colour vision deficiencies) and it is clear, which colour is meant with "orange" we would like to keep the colour description as it is.

Line 475: 16S rRNA gene

Reply: Ok, we will change this accordingly.

Line 475: as mentioned above, please provide details on how these comparisons were made (i.e. were all of the same methods followed? Comparison of reactions efficiencies, same extraction kits, normalization to gene copy number to g of soil, etc.)

Reply: Thanks for the comment. Please find a detailed reply on the method comparison topic below your major comment above.

Line 492: were the ordinations made only on the DNA data? Were ordinations of cDNA data similar? Perhaps these could be presented in the supplemental data if they add to the story?

Reply: Thank you for this idea. Yes, the ordinations were only done for the DNA data, because they are available from all locations at all points in time. We believe that ordinations for the cDNA data would provide no additional insights into time-dependent community changes since cDNA was only obtained from location HC2 in the drought and post-inflow year, not for the baseline conditions and not for any of the other sampling sites. This means, that these ordinations would not enable us to show the effect of the brackish water inflow and the drought, because they cannot be compared to the previous conditions. In addition, there are only 23 data points of cDNA, which we consider to be too little for a meaningful ordination.

Figure 1: I did not find a reference to this figure in the manuscript? It's an excellent figure and should be included.

Reply: Yes, you are right, we should definitely find a place for references. We suggested to include it at the following lines:

l.33: "We found that both, the inflow effect of brackish water and in parts also the preceding drought increased the sulfate availability in the surface and pore water (see Fig.1)."

l. 557: "Therefore, the drought cannot be the only source for the observed increase in pore water ion concentrations and hence, we can assume that both, brackish water inflow and not only the legacy effect of the drought in 2018 changed sulfate concentrations in the surface and pore water and was critical for the methane dynamics and the microbial community composition (Fig. 1)."

l. 607: "If anaerobic $CO_2$ production had been a result of methane oxidation, it had to happen in an area outside the scope of our analysis, namely the water column or the fresh litter layer above the peat soil (Fig. 1)."

l. 653: "As discussed earlier, though, methane oxidation most likely occurred in the standing water above the peat (Fig. 1) given the substantial drop in methane emissions despite the fact that methanogenesis seemingly occurred besides alternative anaerobic pathways of carbon respiration, mostly sulfate reduction."

l. 669: "It remains unresolved, however, why methane emissions decreased to a new minimum since rewetting more than a decade ago, while methanogenic absolute abundances and methane concentrations overall did not change or even decreased (Fig. 1)."

---

## Author Response (AR2)

Response to the final editorial comments:

Dear Denise,

Thank you very much for accepting the manuscript and for your final suggestions and edits. We will go through line by line and reply and your suggestions singly.

l. 27: Thank you for the suggestion. However, we would like to keep the reference for Fig. 1, since it is hard to find good chances to point at this very broad summarizing figure. In addition, Referee #2 suggested to include more references for Fig. 1.

l. 33 and others: Thank you for this hint. We decided to change all "cDNA" to "RNA" except in the lines: 266, 267, 277 and within Fig. 4. Furthermore, we would slightly adapt the caption of Fig. 1 to:

"Figure 4: Bubble plots showing the microbial community composition and relative abundances from all sampling locations along the surface water salinity gradient (a) and the sampling location HC2 (b and c). On the y-axes the taxonomical groups on order (methanogens, methanotrophs), class (sulfate reducing bacteria (SRB)) and genus level (anaerobic methanotrophic archaea (ANME)) are displayed. The x-axes show a) the locations HC1-4 and sampling depths, where codes correspond to the following depths: 1 = 0-5, 2 = 5-20, 3 = 20-40, 4 = 40-50 cm and b) and c) the depth in cm. Coloring reflects the different microorganism groups. Circle sizes represent relative abundances (sqrt transformed) of different taxonomic groups from a, b) DNA- and c) **RNA-based** sequencing **(cDNA data derived from RNA extraction)**. Note, that groups are not adding up globally, but sum up to 100% within each group (methanogens, methanotrophs, SRB, ANME). Please, also note that preservation methods differed slightly between the studies."

l. 72 and others: Thank you for the suggestions. However, we would like to keep the domain names as lowercase.

l. 111: Thanks for paying attention to the introduction of abbreviations. Actually, SRB were introduced in line 69, so line 83 should be remain as it is. But we will remove the sulfate reducing bacteria and only leave the SRB in l. 111.

l. 154: We will delete the comma, thanks for the hint.

l. 221: Thanks, we will close the parenthesis.

l. 233: Ok, we will define ICP-OES as "inductively coupled plasma optical emission spectrometry" and insert ICP-OES into the parenthesis.

l. 255: Thank you very much for this correction. We will change the sentences to: "DNA concentrations were quantified using a Qubit 2.0 Fluorometer (ThermoFisher Scientific, Darmstadt, Germany), following the protocol of the DNA **High Sensitivity** and Broad Range Assay Kit (**dsDNA HS and BR Assay**, ThermoFisher, Berlin, Germany)."

l. 258: We will also adapt this sentence slightly to "RNA concentrations were also quantified with the Qubit 2.0 Fluorometer and the RNA **High Sensitivity** Assay Kit (**RNA HS Assay**, ThermoFisher, Berlin, Germany)."

l. 270: Thank you for the careful read. We will add the missing part and change to: 1 µl 0.1M DTT.

l. 275: Ok, yes, we will include references and add them to the reference list, as well as to the sentence: "Amplification via polymerase chain reaction (PCR) of 16S rRNA genes of DNA and cDNA samples was performed using the universal primer combination Uni515-F/ Uni806-R **(Caporaso et al., 2011)**, for both, bacteria and archaea, and primer combination S-D-Arch-0349-a-S-17/ S-D-Arch-0786-a-A-20 **(Takai and Horikoshi, 2000)** for more precise archaea detection."

l. 279: We agree that the term "backup" was misleading since it suggests redundancy of data. We had intended to say that the sequencing reads obtained with the universal primer provides support for the absolute quantification since the universal primer targets both bacteria and archaea. Therefore, the relative abundance of methanogens based on read counts can be put in context with the mcrA gene copy numbers retrieved through qPCR. We will change the wording in the manuscript to " We decided for the universal primer, because it has equal resolution for bacteria, but covers both, bacteria and archaea providing **some support for the qPCR data**."

l. 296-298: Yes, we will change the sentences accordingly to: "Whereas primers for 16S rRNA **genes** (Eub341-F/Eub534-R) target general prokaryotic microorganisms, primers used to amplify mcrA, pmoA, and drsB are specific **to genes encoding enzymes used by** methanogenic archaea (mcrA, mlas-F/mcrA-R), aerobic methanotrophic bacteria (pmoA, pmoA189-F/pmoA661-R) and **SRB** (dsrB, DsrB2060-F/DsrB4-R)."

l. 303: We will insert "gene" behind 16S rRNA.

l. 304: We will add "gene copy numbers" and delete "of concentrations".

l. 305: Would "genes" not belong behind 16S rRNA to be consisted with the other text? We will insert it as you suggested so long.

l. 306: We will insert "gene" behind 16S rRNA.

l. 307: We will use the abbreviation SRB.

l. 318: Yes, you are right, pre-processing was done on our data only. Therefore, we would like to include the sentence almost at the very end of the paragraph: "The Illumina paired-end (PE) sequences were preprocessed by the method described in Krauze et al. (2021) and Yang et al. (2021). Briefly, demultiplexing was implemented by combining mothur (version 1.39.0) (Schloss et al., 2009), BBTools (Bushnell, 2014) and a custom python script. The PE reads were processed with the 'make.contigs' function of mothur and the resultant report and groups files were parsed with a custom python script to get sequence identifiers of the good quality contigs (minimum overlap length > 25, mismatch bases <5 and without ambiguous base) for each sample. Next, PE sequences were extracted for each sample with the 'filterbyname.sh' function of BBTools. After these steps, orientation of PE sequences was corrected by 'extract_barcodes.py' function of QIIME (version 1.8) (Caporaso et al., 2010). After removing primers with awk command, the PE sequences were fed to DADA2 (Callahan et al., 2016) for filtering, dereplication, chimera check, sequence merge, and amplicon sequence variants (ASV) calling. **All sequencing reads, including those from Wen et al. (2018) and Unger et al. (2021) were merged into a common ASV file which provided the basis for all following analyses.** Taxonomic assignment was referred to SILVA138 (Quast et al., 2013) in platform QIIME2 (Bolyen et al., 2019)."

l. 330: Thank you for the hint. However, we would like to keep the heading in its original state. Also, because in addition to microbial data, pore water data and their analysis are described here (see l. 349).

p.11, below: Thank you for this valuable hint for the future. We shared this among the co-authors and will consider this method in further studies.

l. 424: We will delete "afterwards".

l. 529: We would like to change the sentence to: "Sulfate was however **not a significant variable among** the bacterial nor the archaeal communities."